# The HUSH complex is a gatekeeper of type I interferon through epigenetic regulation of LINE-1s

Hale Tunbak[1,7], Rocio Enriquez-Gasca [1,7], Christopher H. C. Tie[2], Poppy A. Gould[1], Petra Mlcochova[3], Ravindra K. Gupta[3], Liane Fernandes[1], James Holt[1], Annemarthe G. van der Veen[4,6], Evangelos Giampazolias[4], Kathleen H. Burns [5], Pierre V. Maillard [1] & Helen M. Rowe [1✉]

The Human Silencing Hub (HUSH) complex is necessary for epigenetic repression of LINE-1 elements. We show that HUSH-depletion in human cell lines and primary fibroblasts leads to induction of interferon-stimulated genes (ISGs) through JAK/STAT signaling. This effect is mainly attributed to MDA5 and RIG-I sensing of double-stranded RNAs (dsRNAs). This coincides with upregulation of primate-conserved LINE-1s, as well as increased expression of full-length hominid-specific LINE-1s that produce bidirectional RNAs, which may form dsRNA. Notably, LTRs nearby ISGs are derepressed likely rendering these genes more responsive to interferon. LINE-1 shRNAs can abrogate the HUSH-dependent response, while over-expression of an engineered LINE-1 construct activates interferon signaling. Finally, we show that the HUSH component, MPP8 is frequently downregulated in diverse cancers and that its depletion leads to DNA damage. These results suggest that LINE-1s may drive physiological or autoinflammatory responses through dsRNA sensing and gene-regulatory roles and are controlled by the HUSH complex.

[1] Centre for Immunobiology, Blizard Institute, Queen Mary University of London, London E1 2AT, UK. [2] Infection and Immunity, University College London, London WC1E 6BT, UK. [3] Department of Medicine, University of Cambridge, CB2 0AF Cambridge, UK. [4] The Francis Crick Institute, 1 Midland Road, London NW1 1AT, UK. [5] Department of Pathology, John Hopkins University School of Medicine, Baltimore, MD 21205, USA. [6] Present address: Leiden University Medical Centre, Department of Immunohematology and Blood Transfusion, Albinusdreef 2, 2333 ZA Leiden, The Netherlands. [7] These authors contributed equally: Hale Tunbak, Rocio Enriquez-Gasca. ✉email: h.rowe@qmul.ac.uk

Throughout evolutionary history, the human genome has been continuously bombarded with genome invasions, either from exogenous retroviruses that have infected the germline, or from new waves of endogenous transposition[1]. This process has led to up to two-thirds of our genome now consisting of repetitive DNA[2]. Transposition is important to drive genome innovation, and there is an increasing number of documented cases where transposable elements (TEs) have been repurposed, for example to act as enhancers or promoters, with functions in pathogen defense[3]. Regulation of the mobile genome is critical to maintain genome integrity, however, and this is largely achieved through epigenetic silencing. For example, long interspersed element-1s (LINE-1s) are targeted by histone deacetylation, H3K9me3 and DNA methylation[4–7]. Recent data have revealed that inactivation of epigenetic silencing, particularly in human cancer cell lines leads to the reactivation of endogenous retroviruses and other TEs[8–15]. This is accompanied by activation of cytosolic nucleic acid sensors of the innate immune system and production of type I interferons (IFNα and IFNβ, reviewed in ref. [16]). Such activation can overcome resistance to checkpoint blockade and promote anti-tumor immunity in mouse cancer models[14], although type I IFNs can worsen cancer progression and disease outcome when they are constitutively produced[17]. IFN production following treatment of colorectal or ovarian cancer cells with pharmacological inhibitors of DNA methylation appears to depend on double-stranded RNA (dsRNA) sensing through the RIG-I-like receptors (RLRs), MDA5 (melanoma differentiation factor 5) and RIG-I (retinoic acid-inducible gene I) and activation of their downstream mitochondrial adaptor, MAVS (mitochondrial antiviral signaling protein)[8,13]. Very little is understood, however, about which classes of repetitive DNA potentially produce RNA molecules that engage with the RLRs and mediate cross-talk with innate and adaptive immunity, either in physiological or disease contexts. Likewise, little is known about which epigenetic pathways are involved in transcriptional control of repetitive DNA in human adult tissues[12,15] or in cancer cells[10,14]. Notably, however, Alu elements that exhibit an inverted repeat configuration in the genome have been identified to form duplex RNA recognized by the dsRNA sensor MDA5[18–20] and LINE-1 elements are implicated in IFN activation, including through cGAS (cyclic GMP-AMP synthase) and STING-(stimulator of IFN genes)-dependent DNA sensing[21–24].

The human silencing hub (HUSH) complex was discovered as a novel epigenetic complex responsible for position-effect variegation of integrated transgenes in human cells[25]. It is associated with H3K9me3-dense genomic regions, including the 3 prime ends of zinc finger genes and functions to repress zinc finger genes and ribosomal DNAs[25–27]. The HUSH complex partners with the chromatin regulator MORC2, which contains a zinc finger and an ATPase domain[26]. Mutations in MORC2 are associated with neuropathies and these disease mutations have been found to affect MORC2 structure and HUSH function[28]. The HUSH complex has also been discovered to mediate epigenetic silencing of murine leukemia virus through its recruitment by NP220 to the viral DNA before its integration, thereby preventing viral infection[29]. HUSH exerts some position-dependent repression on HIV, which encodes accessory proteins that can counteract HUSH repression, through mediating degradation of HUSH complex components[25,30,31].

We and others have recently demonstrated the HUSH complex to be necessary to repress LINE-1 elements[32–34]. We showed that the HUSH complex represses LINE-1s and co-regulated genes in naïve mouse embryonic stem cells[34] and is mainly required to regulate the youngest species-specific LINE-1s[32,34]. Interestingly, MORC2 exerts some binding specificity for full length LINE-1s[33]. Here, we reasoned that HUSH-regulated LINE-1s with full-length

mRNAs and potentially intact ORFs may have been repurposed to drive nucleic acid sensing in human adult tissues. We show here that the HUSH complex negatively regulates the type I IFN response in human cells through epigenetic regulation of LINE-1 elements. Furthermore, we provide evidence that young LINE-1s produce bidirectional transcripts that could form dsRNA.

## Results

**The HUSH complex regulates type I IFN signaling.** We asked whether the HUSH complex regulates type I IFN induction and responsiveness in human cells by employing shRNA-mediated HUSH depletion in a HEK293 IFN reporter cell line. These cells have been described before[35,36] and harbor an integrated lentivector expressing destabilized GFP under control of IFN-stimulated response elements (ISRE), referred to as ISRE-GFP (Fig. 1a). We verified by western blot or qRT-PCR that HUSH components were depleted by day 4 post introduction of the shRNAs (Fig. 1b) and found that the ISRE-GFP reporter becomes activated at day 6 post shRNA addition (Fig. 1c). While depletion of MPP8 led to a potent IFN response, periphilin (PPHLN1)-depletion induced a significant but lesser response, and TASOR-depleted cells only exhibited a minor shift in ISRE-GFP reporter expression. Controls had no effect on the reporter (Fig. 1c and Supplementary Fig. 1a). Interestingly, combining TASOR and MPP8 depletions revealed that TASOR-depletion could partially inhibit the endogenous IFN-stimulated gene (ISG) response mediated by MPP8-depletion (Fig. 1d). However, induction of the ISRE-GFP reporter was more potent in TASOR-depleted cells compared to control cells each treated with IFN-β (Fig. 1e). This suggests that TASOR-depleted cells are poised for innate immune activation.

**The HUSH complex controls type I IFN induction.** We then employed MPP8-depletion to pursue the mechanism further and first verified that the ISRE-GFP reporter was activated upon MPP8-depletion using additional MPP8 shRNAs (Supplementary Fig. 1a). We confirmed that endogenous ISGs were induced upon MPP8-depletion in three additional human cell lines: HeLa cells, THP-1 cells (acute monocytic leukemia cell line) and untransformed primary foreskin fibroblasts (HFFs) (Fig. 1f and Supplementary Fig. 1b). Supernatant from MPP8-depleted 293 cells was sufficient to activate the ISRE-GFP reporter cells in an IFN bioassay (Fig. 1g) and MPP8-depletion led to an induction of IFN-β itself, shown here by employing an IFN-β reporter cell line in which GFP is driven by an IFN-β promoter[36] and by IFN-β ELISA (Fig. 1h, i). We found that expression of the ISRE-GFP reporter was dependent on JAK/STAT signaling by employing a JAK/STAT inhibitor (Fig. 1j and Supplementary Fig. 1c for the endogenous response). PPHLN1-depletion led to a significant induction of type I IFN and JAK/STAT-dependent ISG expression (Supplementary Fig. 1d–f), albeit to a lesser extent than MPP8-depletion. Finally, an IFN bioassay revealed that IFN is also produced by primary fibroblasts upon MPP8 or PPHLN1-depletion (Fig. 1k). These data suggest that the HUSH complex regulates the induction and response to type I IFNs.

**The HUSH complex regulates genes involved in inflammation.** To understand the global phenotype of HUSH-depletion, we performed total RNA-sequencing of primary fibroblasts. Results for MPP8-depletion showed a clear upregulation of genes involved in immunity and inflammation (Fig. 2a, b, dataset 1 in the Source data file), including 204 ISGs, which typically function in innate and adaptive immunity (employing a list we defined before in ref. [15]). Of the fraction that were not defined as ISGs, 16 were KZNFs in accordance with the known function of the

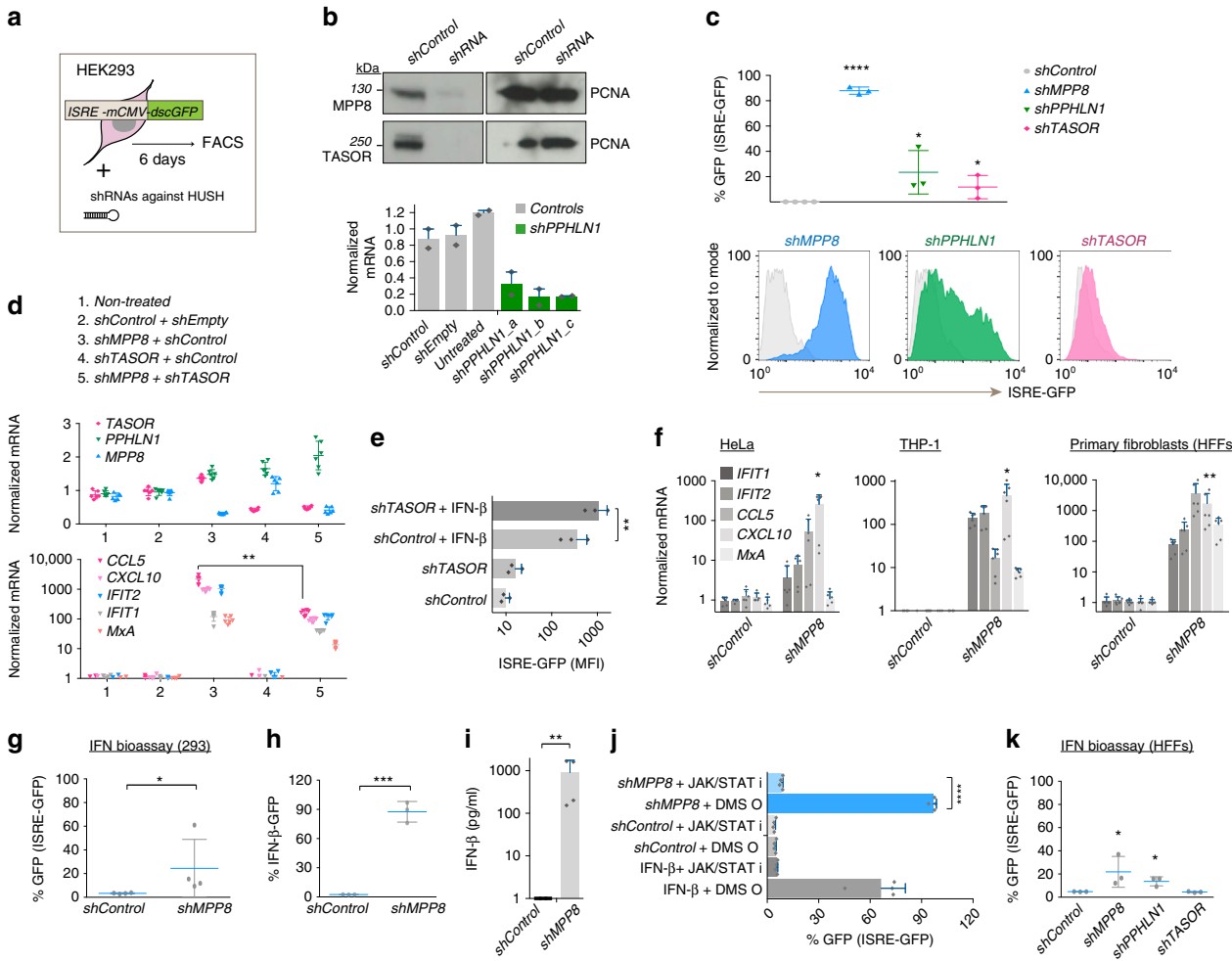

**Fig. 1 The HUSH complex regulates the type I IFN response. a** Diagram of IFN induction assay: HEK293 cells harbor an integrated IFN reporter expressing destabilized GFP (dscGFP). ISRE: IFN-stimulated response elements; *mCMV*: minimal CMV promoter. GFP was assessed 6 days post shRNAs. **b** Western blot and qRT-PCR to assess HUSH depletion day 4 post shRNAs. *PPHLN1* mRNA normalized to *GAPDH*. $N = 3$ biologically independent samples with technical qRT-PCR duplicates shown. **c** ISRE-GFP induction 6 days post shRNAs. $N = 3$ biologically independent experiments. Two-tailed unpaired *t* tests of shRNAs vs. *shControl*: $p = <0.0001$ (shMPP8), $p = 0.0372$ (shPPHLN1) and $p = 0.0466$ (TASOR). **d** shMPP8 was combined with *shTASOR* or controls as stated. qRT-PCR of endogenous ISG mRNA expression (*GAPDH* normalized). $N = 3$ biologically independent experiments with technical duplicates shown. A two-tailed unpaired *t* test was used to compare *shMPP8 + shControl* to *shMPP8 + shTASOR* for *CCL5* ($p = 0.0021$). **e** *shTASOR* or *shControl-treated* cells were treated with IFN-β and ISRE-GFP measured 24 h later. $N = 3$ biologically independent experiments. *P* value: 0.0037 (two-tailed paired *t* test). MFI: mean fluorescent intensity. **f** QRT-PCR of ISG expression (*GAPDH* normalization) 6 days post shRNAs (or 3 days in the case of HeLa cells). $N = 3$ biologically independent experiments with technical duplicates shown. Two-tailed paired *t* tests were used to compare controls to *shMPP8* for CXCL10: $p = 0.0473$ (HeLa), $p = 0.0229$ (THP-1s) and $p = 0.0084$ (HFFs). **g** IFN bioassay: supernatants from *shRNA*-treated 293 cells were added to ISRE-GFP reporter 293 cells and GFP expression measured 24 h later. $N = 4$ biologically independent samples. $P = 0.0240$ (two-tailed paired *t* test). **h** 293 cells containing an IFN-β-destabilized GFP reporter were transduced with the stated shRNAs and GFP measured 6 days later. $N = 3$ biologically independent samples. $P = 0.0005$ (two-tailed paired *t* test). **i** Supernatants were harvested 6 days post *shRNAs* and used for an IFN-β ELISA. $N = 4$ biologically independent samples. $P = 0.0023$ (two-tailed paired *t* test). **j** 293 cells treated with *shRNAs* in the presence of the JAK/STAT inhibitor Ruxolitinib or DMSO (added from day 1) were harvested for GFP FACS at day 6. $N = 4$ biologically independent samples. IFN-β ± Ruxolitinib served as a positive control. $P = <0.0001$ (two-tailed unpaired *t* test). **k** IFN bioassay: supernatants from *shRNA*-treated HFFs were added to ISRE-GFP reporter 293 cells and GFP expression measured 24 h later. $N = 3$ biologically independent samples. Two-tailed paired *t* test *p* values: $p = 0.0469$ (shMPP8), $p = 0.0320$ (shPPHLN1). All data presented in this figure show mean values ± SD except (**b**), which depicts mean values ± SEM.

HUSH complex in binding to and regulating KZNF genes[25,26,33] and most of these were previously identified to be MORC2-bound[33] (Fig. 2b and Supplementary Fig. 2). Other upregulated genes were enriched for roles in chromatin regulation, cell cycle and DNA damage (Supplementary Fig. 3a–c). Of note, the IFN and inflammatory gene signature upon MPP8-depletion was reproduced in HeLa cells (Supplementary Fig. 3d). We used gene set enrichment analyses to assess the phenotype observed upon

depletion of individual HUSH components in primary fibroblasts (Fig. 2c, dataset 1 in the Source data file). TASOR-depleted samples differed from MPP8 and PPHLN1 in their lack of an activated IFN signature but overlapped in terms of TNF-α signaling and G2M cell cycle checkpoint. This might indicate that greater depletion of TASOR than that achieved here was required to observe a more complete phenotype. Concordance between PPHLN1- and MPP8-repressed genes was evidenced by a large

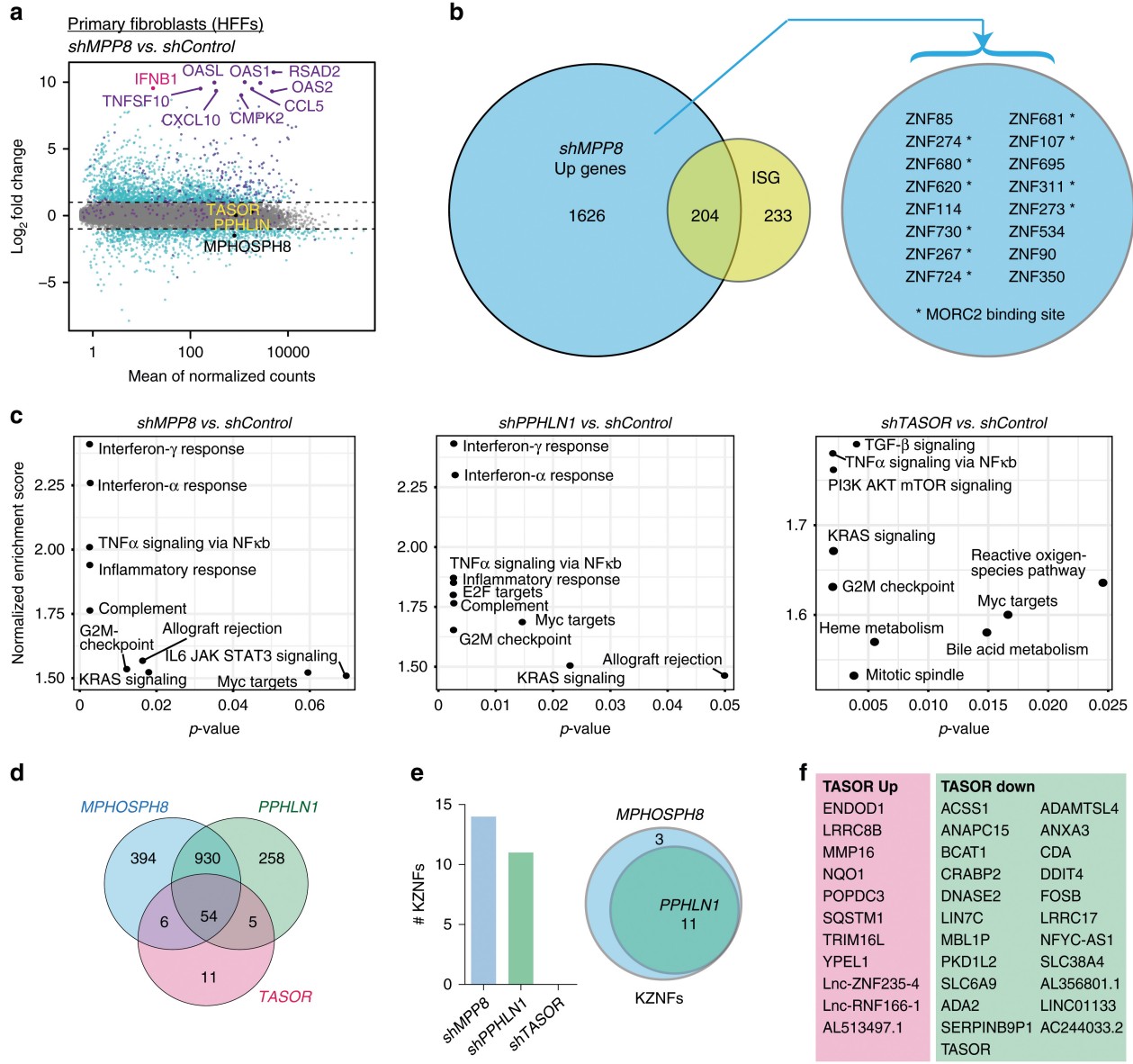

**Fig. 2 The HUSH complex regulates genes involved in inflammation.** Primary fibroblasts (HFFs) were treated with *shRNA* vectors, and RNA harvested 6 days later for total RNA-sequencing. *N* = 6 biologically independent experiments. **a** MA plot showing results of differential expression analysis, where significantly differentially expressed genes (log2 fold change >1 and a *p*-adjusted value < 0.05 after Benjamini–Hochberg multiple testing correction of Wald test *p*-value of shMPP8/shControl) have been highlighted in dark cyan and ISGs are shown in purple. See dataset 1 in Source data for all exact *p* values. Example upregulated ISGs are labeled as well as HUSH components (of which MPHOSPH8/MPP8 is downregulated). **b** Venn diagram showing overlap of ISGs (list defined in ref. [15]) with genes upregulated in MPP8-depleted samples. Upregulated KZNFs are listed. See also Supplementary Fig. 2 for MORC2 binding data using public ChIP data[33]. **c** Total RNA-seq data were generated for samples depleted for each HUSH component vs. shControls (*n* = 3 biologically independent experiments and data are represented as gene set enrichment analyses using fgsea, where nominal p-values were calculated for 1000 permutations and corrected for multiple-testing using the Benjamini–Hochberg method). **d** Venn diagram showing overlap of genes upregulated in each HUSH-depletion. **e** Comparison of number and overlap of KZNFs upregulated in HUSH-depleted cells. **f** Significantly differentially-expressed genes unique to TASOR-depleted samples are listed.

number of mutually-upregulated genes (Fig. 2d), which included KZNFs for MPP8 and PPHLN1 but not TASOR (Fig. 2e). Genes uniquely up or downregulated in TASOR-depleted cells are depicted in Fig. 2f and interestingly, several of these (SQSTM1 and ACSS1) have immunomodulatory roles[37,38].

**MPP8-depletion is accompanied by RLR-dependent RNA sensing.** There is mounting evidence that epigenetic pathways prevent TEs from nucleic acid sensing through RLRs, particularly

in cancer cells[8,10,13,14]. We detected expression of components of the RLR-signaling pathway, MDA5, RIG-I, and MAVS in all cell lines in which we had observed an IFN response upon HUSH-depletion (Fig. 3a). cGAS, in contrast, was not detected in 293 cells following IFN-β treatment, although it was readily expressed in THP-1 cells in which it was upregulated upon IFN-β treatment (Fig. 3a). We confirmed that the IFN response was MAVS-dependent using shRNA depletion of MAVS in ISRE-GFP and IFN-β GFP 293 reporter cells and we verified that MAVS-depleted cells could no longer react to stimulation with poly(I:C),

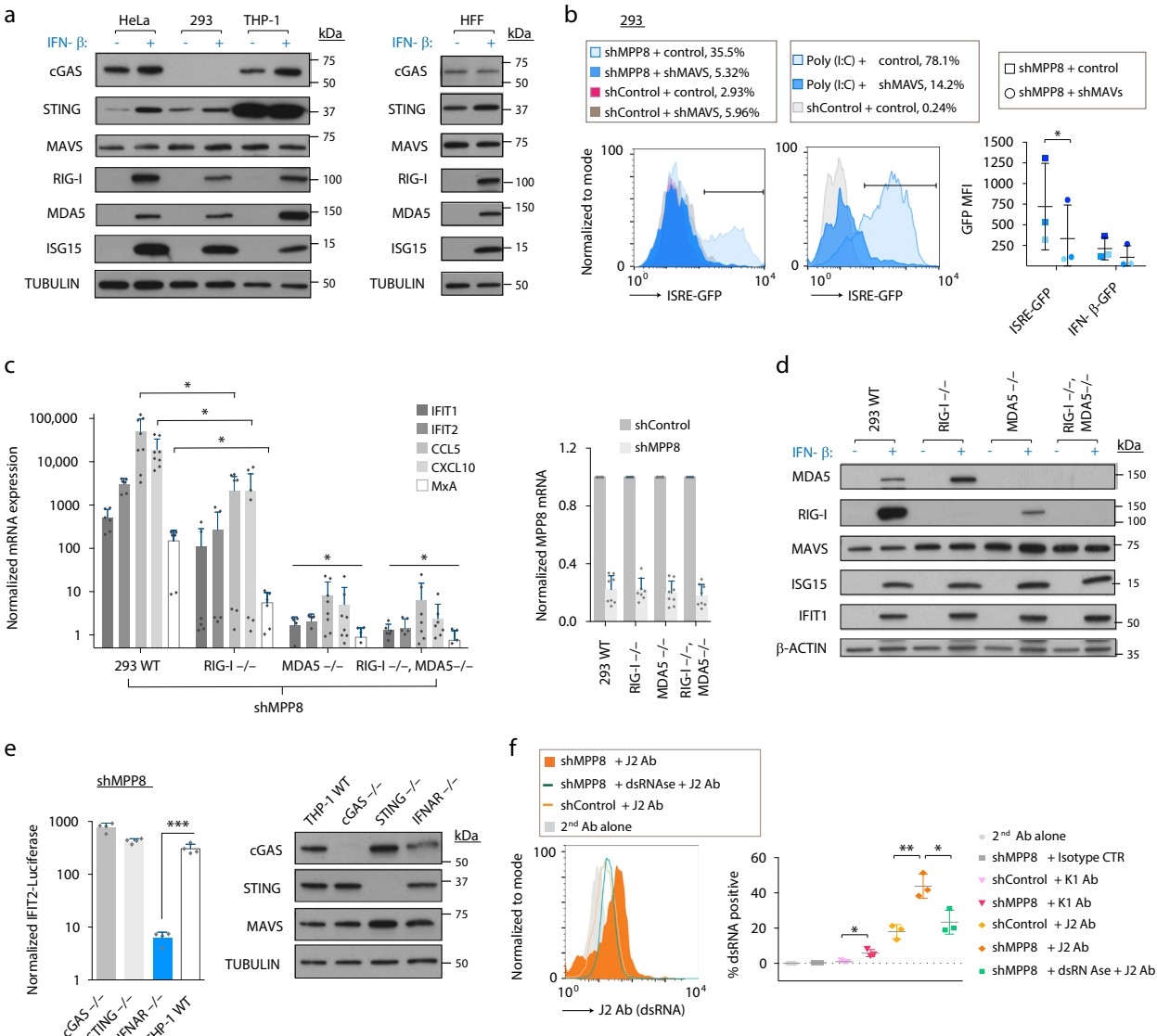

**Fig. 3 MPP8-depletion is accompanied by RLR-dependent RNA sensing. a** Western blots on extracts from stated cells treated ± IFN-β for 24 h. $N = 2$ independent blots with one representative blot shown. **b** 293 cells were co-transduced with *shControl* or *shMPP8* plus *shMAVS* or a control vector of the same backbone (*hygromycin*) and GFP was measured on day 6 (left). MAVS depletion was verified using polyI:C, added 24 h before FACS (middle). Summary data shown for experiments in ISRE-GFP and IFN-β-GFP reporter cells (right, $n = 3$ biologically independent samples). $P = 0.0372$ (one-tailed paired $t$ test). MFI: mean fluorescent intensity. **c** Left: qRT-PCR expression of endogenous ISGs in the stated shRNA-treated 293 cell lines at day 6 (*GAPDH normalized*). $N = 4$ biological independent experiments with technical duplicates shown. CCL5 and CXCL10 were measured in all experiments while the other 3 ISGs were measured in 3 of the experiments. One-tailed paired $t$ tests showed expression of all ISGs was significantly lower in MDA5−/− cells and double knockout (DKO) cells than in wildtype (WT) controls. IFIT1: $p = 0.0063$ (MDA5−/−), $p = 0.0026$ (DKO); IFIT2: 0.0007 (MDA5−/−), 0.0019 (DKO); CCL5: 0.0002 (MDA5−/−), 0.0028 (DKO); CXCL10: 0.0001 (MDA5−/−), 0.0007 (DKO); MxA: 0.0427 (MDA5−/−), 0.0423 (DKO). Expression of CCL5, CXCL10, and MxA was significantly decreased in RIG-I−/− cells compared to controls. CCL5: $p = 0.0183$; CXCL10: $p = 0.0372$; MxA: $p = 0.0139$. Right: MPP8 depletion efficiency summary ($n = 4$ biologically independent experiments with technical duplicates shown. **d** Western blots on extracts from cell lines from **c** treated ± IFN-β for 24 h. $N = 2$ independent blots with one representative blot shown. **e** Left: stated knockout or control (WT) THP-1 reporter cell lines were treated with *shRNAs* and assayed for secreted Lucia *luciferase* (driven by IFIT-2) 6 days later. $N = 4$ biologically independent experiments. *Luciferase* readings were normalized to background values in *shControl*-treated cells. $P = 0.0003$ (one-tailed paired $t$ test). Right: western blots on extracts from stated cells. $N = 3$ independent blots with one representative blot shown. **f** 293 cells were treated with *shRNAs* and fixed 6 days later for intracellular staining with dsRNA antibodies, J2/K1 or control antibodies. $N = 3$ biologically independent samples. Two-tailed unpaired $t$ test $p$ values = 0.0166 (shMPP8 vs. shControl for K1 Ab); 0.0048 (shMPP8 vs. shControl for J2 Ab); 0.0212 (shMPP8 + J2 Ab ± dsRNase). All data presented in this figure show mean values ± SD.

a dsRNA analog (Fig. 3b). To dissect whether MAVS signaling was mainly dependent on RIG-I or MDA5, we employed CRISPR/Cas9-generated knockout 293 cells for either or both dsRNA sensors. Results showed that the response is dependent on

MDA5 with a contribution from RIG-I (Fig. 3c). We verified by western blot that expression of MDA5 and RIG-I were detected in the expected cells upon their treatment with IFN-β (Fig. 3d). Note that MDA5 is more highly expressed in RIG-I knockout cells than

control cells, potentially to compensate for RIG-I loss, while in MDA5 knockout cells, RIG-I expression is low, possibly contributing to the low IFN response in these cells upon MPP8-depletion (Fig. 3d). Importantly, we verified that all wildtype and knockout cell lines could mount a response to IFN-β treatment by measuring the induction of IFIT1 and ISG15 (Fig. 3d).

We then asked if the cGAS/STING pathway was necessary for IFN signaling upon MPP8-depletion in a cell line efficient for DNA sensing, (Fig. 3a). For this, we employed commercially-available THP-1 cells that stably express an IFIT2-responsive *luciferase* reporter that are either knockouts for the DNA sensors cGAS or STING, or for the IFN type-I receptor (IFNAR). The observed IFN response was independent of cGAS/STING but dependent on signaling through the type I IFN receptor, as expected (Fig. 3e). Knockout of DNA sensors was verified by western blot and functionally by stimulation with agonists (Fig. 3e right and Supplementary Fig. 3e). Since results suggested that the IFN response is dependent on dsRNA (Fig. 3b–d), we tested whether we could detect dsRNA in MPP8-depleted cells by employing antibodies (J2 and K1) that recognize dsRNA. DsRNA was apparent in MPP8-depleted cells and the signal was abrogated by pretreatment with the long dsRNA-specific endoribonuclease, RNase III (Fig. 3f).

**MPP8-depletion results in overexpression of LINE-1s and LTRs.** We and others previously showed that the HUSH complex mediates epigenetic repression of LINE-1 elements[32–34], and we hypothesized that LINE-1 RNAs could be driving the IFN response in MPP8-depleted cells. However, the HUSH complex also represses MLV[29] and potentially also HIV[25,30,31]. We, therefore, first verified that our shRNA lentivector itself was not necessary to drive the ISG response, by using siRNAs against MPP8 to show that there is still induction of the ISRE-GFP reporter (Fig. 4a). This suggested that the ligand driving the response is endogenous.

We then verified that LINE-1s are overexpressed in MPP8-depleted cells by LINE-1 ORF1-protein western (Fig. 4b). We employed our RNA-sequencing data to determine which TE subfamilies are differentially expressed in MPP8-depleted cells (using TEtranscripts: TEcount tool[39]). We found mainly primate-specific LTR (long terminal repeat) families of between 20 and 50 million years old to be derepressed, including ERV9 and HERVH (Fig. 4c and Supplementary Fig. 4a, b, dataset 2 in the Source data file). We then mapped which precise TE loci were affected in the human genome by assigning uniquely-mapping reads to the genome and intersecting results with coordinates of TEs (using RepeatMasker annotation) and we found mainly LINE-1s, SINEs, and LTRs were upregulated (Fig. 4d, e and Supplementary Fig. 4b, dataset 3 in the Source data file). Although numerous LINE-1 loci were overexpressed, they were not significantly upregulated at the subfamily level using TEtranscripts (dataset 2 in the Source data file). This suggests that LINE-1 upregulation is context-dependent, consistent with the known role of HUSH in regulating TEs in introns of genes[33,34]. Employing available HUSH-binding data revealed that LINE-1s rather than LTRs were likely direct binding targets (Fig. 4d, e and Supplementary Fig. 4b). We highlighted the top five LINE-1 subfamilies in orange containing the most upregulated and most HUSH-bound copies, respectively (Fig. 4d, top and bottom), which spanned families between 12 and 100 million years old.

We next looked at the distance of upregulated LINE-1s and upregulated LTRs to the transcriptional start sites (TSS) of upregulated ISGs, hypothesizing that these TEs may directly regulate these genes (Fig. 4f). The LTRs were significantly closer to ISGs than to random genes suggesting that they may have been

co-opted to act as promoters or enhancers for these genes, thereby contributing to the increased IFN-responsiveness in HUSH-depleted cells (Fig. 1). The LINE-1s were not significantly close to the TSS of ISGs but were enriched within long noncoding RNAs (Supplementary Fig. 4c).

**L1PA1 and L1PA2 are a potential source of dsRNA.** We and others have previously shown that HUSH represses young species-specific LINE-1s, albeit mainly in early development[32–34]. We, therefore, focused on the hominid-specific families, L1PA1 (also called L1HS) and L1PA2, which we noticed were expressed bidirectionally and are known to be MORC2-bound at the 5′ UTR[33]. While L1PA1 is difficult to map at the locus level, due to its high copy number and integrants being mostly identical, we mapped RNA-sequencing reads to the L1PA1 consensus sequence and found that there is significantly more production of sense and antisense strands in MPP8-depleted cells (Fig. 4g). This was also the case for L1PA2, which we show is bidirectionally-expressed as well (Supplementary Fig. 4d). Importantly, in an unbiased approach, we also identified regions in the genome with the highest (top 15%) overlapping expression in both strands and found that L1PA1 was the most overrepresented TE in those regions, followed by L1PA2. Bidirectional LINE-1s (199 L1PA1 and 74 L1PA2) were full-length with a median length of around 6 kb (Supplementary Fig. 4e) and could potentially pair in *cis* or *trans* to produce dsRNAs to accumulate in the cytoplasm. Example bidirectionally-transcribed LINE-1s and upregulated primate-conserved LINE-1s are shown (Fig. 4h, i, Supplementary Fig. 5). Of note, bidirectional L1PA1 loci were 100% identical to the ORF1 protein probed for by western blot, suggesting that we are detecting L1PA1 expression by western (Fig. 4b, dataset 4 in the Source data file).

**LINE-1s regulate the type I IFN response.** Data indicated that the striking IFN response observed upon MPP8-depletion could result from a mixture of components, including LINE-1 RNAs, effects of ZNF transcription factors and unveiled LTRs (Figs. 2, 4 and Supplementary Fig. 6a). We focused on the role of LINE-1 RNAs: We first determined that reverse transcription was not necessary to drive IFN by using reverse transcriptase inhibitors that are effective against LINE-1 RNA and HIV, used here as a positive control (Supplementary Fig. 6b). In contrast, over-expression of ZNF91 and to a lesser extent ZNF93, which are transcriptional repressors of SVA and LINE-1 (L1PA4) elements, respectively[40] and their co-repressor, KAP1 could partially reduce the IFN response (Supplementary Fig. 6c). The modest effect of ZNF93 might relate to the fact that it does not target young LINE-1s, which may be the main source of dsRNA (Fig. 4g and Supplementary Fig. 4d). We, therefore, designed a series of LINE-1 shRNAs and tested them to ascertain if any could inhibit the HUSH-dependent IFN response. We found that two hairpins could partially block the ISRE-GFP reporter response, one designed on the 5'UTR of L1PA1 using the consensus sequence, and one designed on L1PA4 ORF2 using an MPP8-bound element[33] (see Fig. 5a). Additionally, we designed a hairpin against MOV10, an RNA helicase thought to sequester LINE-1 RNA away from MDA5[24,41]. This hairpin led to an augmentation of the IFN response, further suggesting that endogenous LINE-1 RNAs can mediate MDA5-sensing (Fig. 5a). We verified that the LINE-1 shRNAs could also block the endogenous ISG response (Fig. 5b). The shL1PA4 hairpin induced a detectable decrease in global expression of L1PA2-4 RNA by qRT-PCR (Fig. 5c). The L1PA1-2 hairpin recognized L1PA1s and L1PA2s equally (584 LINE-1 integrants, of which 288 were L1PA1 and 271 were L1PA2). Using primers that could detect 76 L1PA2s targeted by

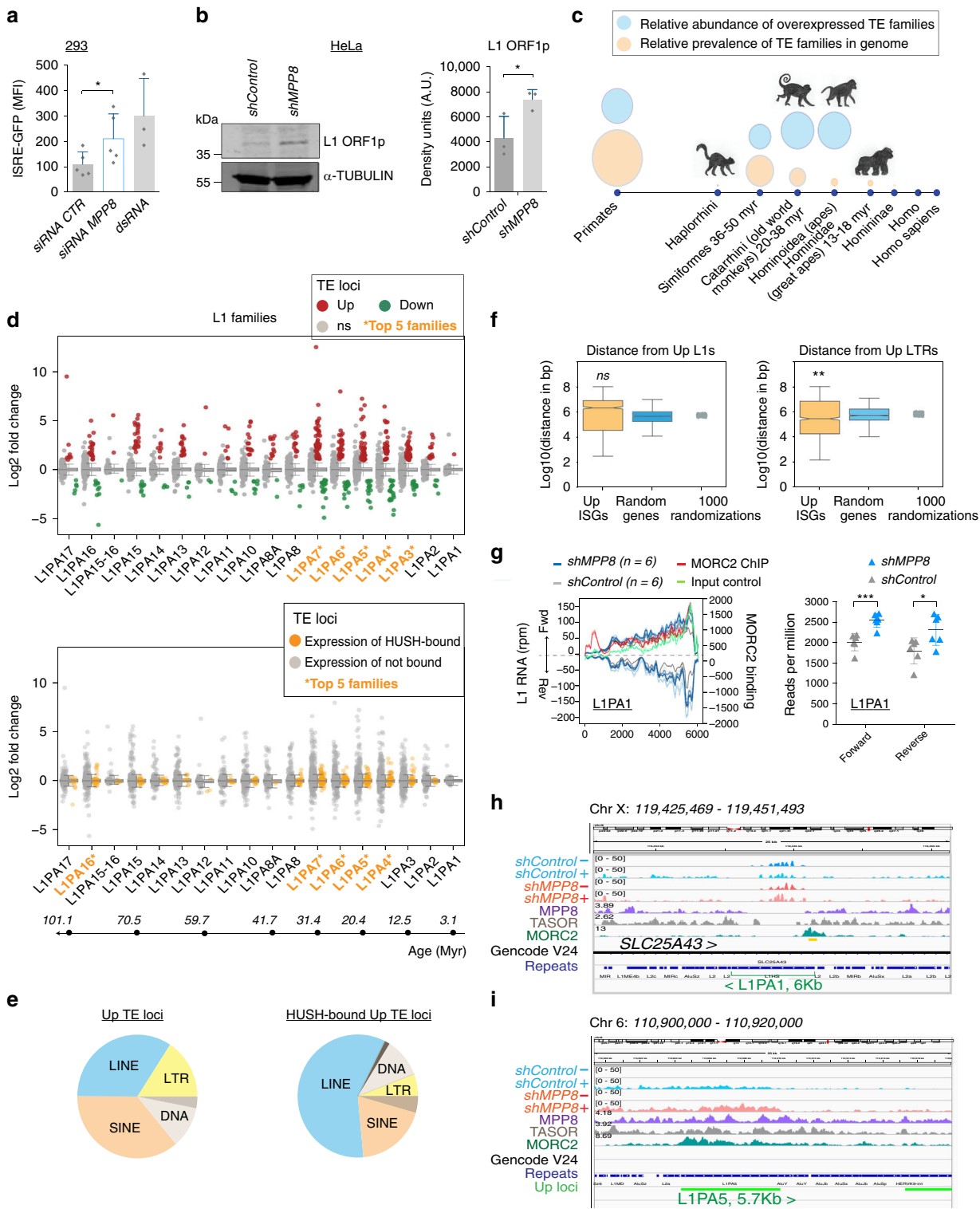

this hairpin, we could show a decrease in LINE-1 upregulation upon MPP8-depletion (Supplementary Fig. 6d).

We next employed public RNA-sequencing data of genes induced upon expression of an engineered full-length active L1PA1 construct (in RPE cells)[21] to see if there was any concordance with genes upregulated in our MPP8-depleted HFF cells. The L1PA1 construct, which was codon-optimized, was doxycycline-induced for 3 days and the response was compared to an induced *luciferase* control to obtain a list of L1-induced genes[21]. There was an overlap of 114 genes, which included 49 ISGs. This was highly significant when compared to the overlap of randomized gene sets of the same number as the number of MPP8-repressed genes (Fig. 5d). Finally, we could show that transfection of an engineered full-length L1PA1 construct was sufficient for induction of the ISRE-GFP reporter in 293 cells (Fig. 5e). These data taken together implicate HUSH-regulated LINE-1 elements in driving type I IFN.

**Fig. 4 MPP8-depletion results in overexpression of LINE-1s and LTRs. a** ISRE-GFP reporter 293 cells were transfected with control (CTR) or MPP8 siRNAs or dsRNA as a positive control and GFP measured on day 6. $N = 5$ biologically independent samples or $N = 3$ for the positive control. $P = 0.0108$ (two-tailed paired $t$ test). Data show mean ± SD. MFI: mean fluorescent intensity. **b** Left: HeLa cells were harvested 4 days post shRNAs and used for western blot for L1 ORF1 protein and alpha-Tubulin. Right: western quantification. $N = 3$ biologically independent samples. $P = 0.0426$ (one-tailed paired $t$ test). Data show mean ± SD. **c** Bubble plots showing the relative prevalence of all TEs in the genome by clade (orange) and relative abundance of overexpressed TE families in *shMPP8* samples compared to controls (blue) using RNA-seq data from Fig. 2 and the software TEtranscripts. See Supplementary Fig. 4a for complete diagram. Monkey images drawn by lab members. **d** Upper panel: TE loci up or downregulated in MPP8-depleted cells (log2 fold change >1, and $p$ adjusted values <0.05 after Benjamini–Hochberg multiple testing correction of Wald test $p$-value of shMPP8/shControl from DESeq2). See dataset 3 in Source data for all exact $p$ values. Results are shown for loci within the stated LINE-1 families. Ns = expression not significantly changed. Lower panel: strip plot showing HUSH-bound (orange) vs. not known to be bound (gray) LINE-1 loci with their relative expression in *shMPP8* samples compared to controls. HUSH peaks defined using epic2, see Supplementary Fig. 3a, using data from ref. [33]. The approximate ages of these LINE-1 families are given underneath using[79]. See Supplementary Fig. 4b for LTR data. The subfamily names with the highest number of upregulated loci (top) or bound loci (bottom) are highlighted in orange. Boxes represent 1st and 3rd quartile, where the central line corresponds to the median; whiskers are ×1.5 of the interquartile range. **e** Pie charts showing upregulated TE loci (left) and HUSH-bound TE loci (right) classified by family. **f** Median distance of Upregulated L1s ($n = 1494$) or LTRs ($n = 1177$) to transcription start sites of upregulated ISGs ($n = 204$) compared to the same number of random genes or to the median distance across 1000 randomizations. The median distance for LTRs is significantly closer than randomized genes ($p = 0.001$, where a one-sided $p$ value was calculated as (1+ no. of randomizations with a median greater than the observed value)/1000 randomizations). Boxplots as in (**d**) where notches indicate the confidence interval. **g** Left: the L1PA1 consensus sequence (repBase) was mapped with RNA-seq reads in both strands (left $y$ axis) or with MORC2 ChIP and total input reads from public data[33] (right $y$ axis). Right: The number of RNA-seq reads mapping forward or reverse strands of the L1PA1 consensus sequence were plotted for all samples. Two-tailed unpaired $t$ test values: $p = 0.0006$ (forward) and $p = 0.0291$ (reverse). Data show mean ± SD. **h** Bigwig visualization of example HUSH-bound young LINE-1 expressed bidirectionally and a HUSH-sensitive primate-conserved LINE-1 expressed in sense (**i**). RNA-seq track scales were all set at 50 and normalized tracks of HUSH-binding were made using data from ref. [33].

---

**MPP8 is frequently downregulated in cancers**. Given the emerging role of epigenetic regulators as therapeutic targets in cancer[42,43], we tested if MPP8 is differentially expressed in the context of this group of diseases. For this, we used the human cancer genome atlas (TCGA)[44] data, focusing on tissues for which there is also available expression data in healthy control samples from GTEx[45], as well as matched TCGA controls. MPP8 was significantly downregulated in 10 out of 15 diverse human cancers as compared to both controls (Fig. 6a). We did not see the same trend for TASOR or PPHLN1 (Supplementary Fig. 6e), which may relate to the more striking phenotype observed of MPP8 depletion (Figs. 1 and 2). To gain insight into whether MPP8 depletion in cancers is associated with an IFN signature, we divided cancers by recently-defined immune subtype categories[17], plotting MPP8 expression levels. We found that MPP8 was most significantly downregulated in cancers with an IFN-γ dominant C2 immune subtype (Fig. 6b). This included high expression of the ISGs, CCL5 and CXCL10.

Given the observed trend across most cancer types to downregulate MPP8, we considered that one of the selective advantages this could confer could be a decreased capacity to repair DNA, also hinted at by the secondary genes upregulated upon MPP8 depletion, which are enriched for DNA damage, cell cycle and chromatin maintenance pathways (Supplementary Fig. 3a–c). This, together with the known risk of active LINE-1 elements in inducing DNA damage and compromising DNA repair[21,46–48], led us to examine if depleting MPP8 in 293 cells was sufficient to lead to DNA damage. We observed that at baseline, MPP8-depleted cells display more damage compared to control cells, as measured by γH2AX staining, and are also compromised in DNA repair following etoposide-induced damage (Fig. 6c). We observed the same trend in HeLa cells (Supplementary Fig. 6f).

## Discussion
Here we reveal that the HUSH complex controls type I IFN signaling in human cells and that the mechanism involves dsRNA sensing by MDA5 and RIG-I. We link this cell-intrinsic innate immune response partly to LINE-1 elements, which become derepressed at the RNA and protein level following HUSH-depletion. Surprisingly, we find that full-length hominid-specific LINE-1 elements, including L1PA1 and L1PA2 produce bidirectional transcripts. This suggests that young LINE-1s are a natural source of dsRNAs of 6Kb in length, ideal candidates for recognition by the long dsRNA sensor, MDA5, which detects dsRNA greater than 2 kb in length[49]. These RNAs were transcribed in control cells and increased in abundance in MPP8-depleted cells. We speculate that these LINE-1 dsRNAs may be chromatin-tethered in control cells and only become cytoplasmic and available for MDA5-sensing upon HUSH-depletion. Of note, we have focused on LINE1s in this study but there are likely many other potential sources of dsRNAs.

Of interest, we identify an increase in LINE-1 mRNAs derived from ancient primate-conserved LINE-1 loci in MPP8-depleted cells. These conserved LINE-1s may have evolved beneficial roles in host gene regulation since many are positioned within introns of genes as noted before[33,34]. Interestingly, these conserved LINE-1s are also enriched in long non-coding RNAs, which can derive from LINE-1s[50]. LINE-1 elements originated more than 160 million years ago[51] and have co-evolved with mammals to regulate genes: LINE-1 and LINE-2 elements have been repurposed in the formation of chimeric transcripts driven by LINE-1 antisense promoters[52], in chromatin-associated non-coding RNAs[53] and in exonization events, including into immunoregulatory molecules[54].

While the potent type I IFN response observed upon MPP8-depletion is likely to be multifactorial, as described for the immune response evoked upon treatment with pharmacological inhibitors of DNA methylation[42], here, we could implicate LINE-1 RNAs and potentially LINE-1-containing transcripts to play a causative role using LINE-1 knockdown and overexpression experiments (see Fig. 7 for a proposed summary model). Importantly, we also found LTRs nearby ISGs to become derepressed upon HUSH-depletion suggesting that the HUSH complex may control type I IFN responsiveness, as well as IFN induction. Of note, ERV9 LTRs that we observed to be derepressed have been described to act as cryptic promoters under epigenetic regulation[55]. LTRs have also been shown to drive production of novel cancer-specific antigens[56].

Finally, we note that MPP8 is frequently downregulated in cancers, which may relate to the increase in DNA damage that we

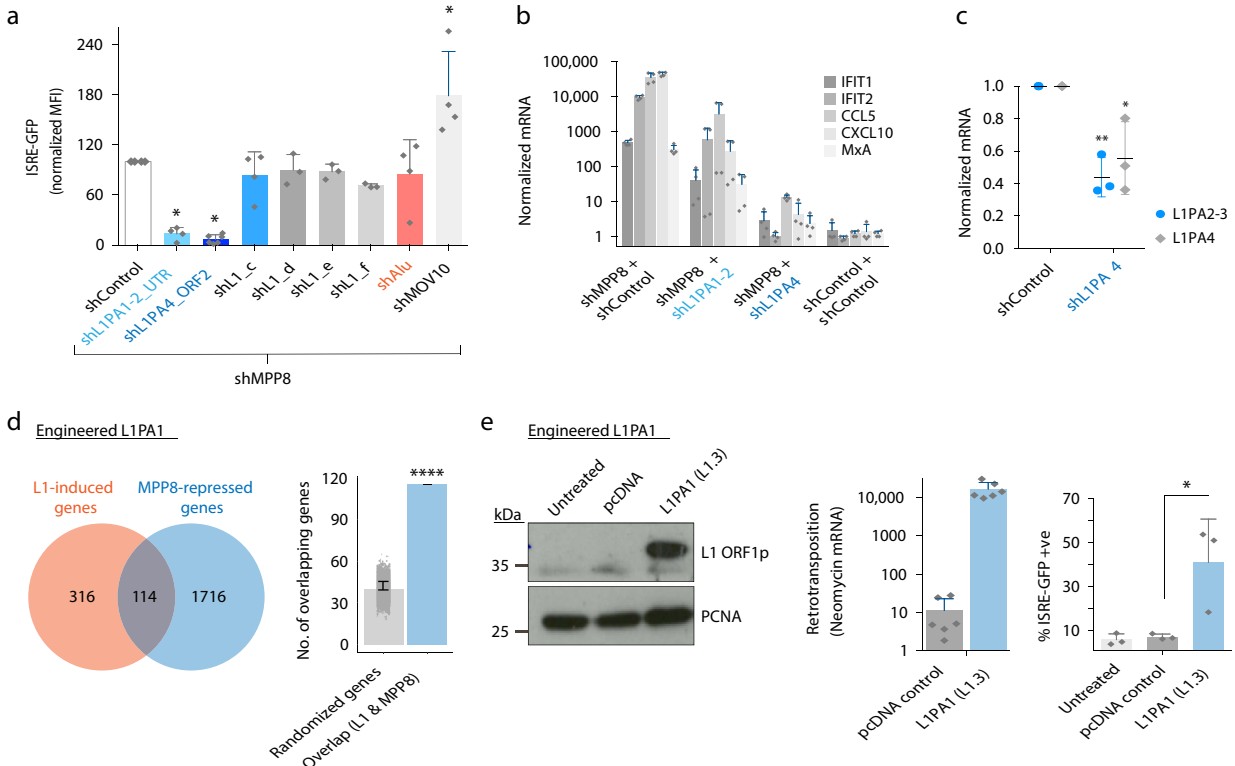

**Fig. 5 LINE-1s regulate the type I IFN response. a** 293 reporter cells were *shControl/shMPP8* treated together with the stated shRNAs against TE families or *shControl* again. GFP MFI (mean fluorescence intensity) at day 6 was normalized to the *shMPP8 + shControl* sample that was set at 100%. N = 4 biologically independent samples, except for shL1_d-f, where n = 3 and shL1PA4, where n = 6. The control was significantly different to the shL1PA1 group (p = 0.0172), the shL1PA4 group (0.0110) and the MOV10 group (p = 0.0265) (two-tailed paired t tests). **b** Groups were setup as in (**a**) but here endogenous ISG expression was measured by qRT-PCR. N = 2 biologically independent experiments with technical duplicates shown. **c** Knockdown efficiency was verified by qRT-PCR using global LINE-1 primers. N = 3 biologically independent experiments. A decrease in global LINE-1 expression was detectable for shL1PA4: p = 0.0013 (L1PA2-3) and p = 0.0268 (L1PA4) (two-tailed unpaired t tests). **d** Venn diagram showing overlap of genes induced by overexpression of an engineered L1PA1 reporter in RPE cells from ref. [21] (a Tet-inducible codon-optimised L1PA1 vs. Luciferase control that was induced for 3 days) and genes upregulated upon MPP8-depletion here. Of 114 genes that overlap, 49 are ISGs. Right: significance was assessed by selecting 10,000 randomizations of the same number (1830) of random genes and looking at the mean number that are shared with the L1-induced genes. P = 5.1203e−24 (hypergeometric test). **e** 293 reporter cells transfected with a pcDNA control or engineered L1PA1 reporter plasmid (ksCMV-101: L1PA1 subtype L1.3). Transfection and retrotransposition verified by western blot and qRT-PCR for spliced Neomycin expression at day 4 (left and middle). N = 3 biologically independent experiments with technical duplicates shown. Right: IFN reporter induction was measured by GFP FACS. N = 3 biologically independent experiments. P = 0.0332 (one-tailed paired t test). All data presented in this figure show mean values ± SD.

observe in MPP8-depleted cells. Whether DNA damage is related to IFN induction[57] or caused by DNA breaks mediated by LINE-1-encoded endonuclease[47] warrants future investigation. It will also be interesting to determine whether MPP8-depletion contributes to carcinogenesis in vivo. Our results fit with a more general emerging role for LINE-1 elements in driving IFNs and inflammation[21–24,58–60]. Of note, although the cGAS/STING pathway was not necessary for the IFN response here, it may contribute to IFN release in vivo and could be upregulated and more active in HUSH-depleted cells. Open questions include uncovering exactly how LINE-1 elements are transcribed bidirectionally and why, and determining whether LINE-1 dsRNA is tethered to chromatin in control cells to mask if from MDA5 sensing. It's not clear if LINE-1 proteins play a role in nucleic acid sensing of LINE-1 dsRNA. Little is known too about how co-opted conserved LINE-1s and other TEs potentially control innate immune genes through gene-regulatory roles and if the HUSH complex is downregulated during physiological immune responses. It is tempting to speculate that LINE-1 elements may play a causative role in driving autoinflammatory diseases and may have been repurposed long ago to naturally prime or potentiate MDA5-mediated innate immunity.

## Methods

**Cell culture and reagents.** HEK293 WT (referred to as 293) or CRISPR/Cas9-generated knockout 293 cells for either RIG-I, MDA5 or both and HeLa cells were cultured in Dulbecco's modified Eagle's medium (DMEM; Gibco, Thermo Fisher Scientific) supplemented with 100 U/ml penicillin/streptomycin (Gibco, Thermo Fisher Scientific) and 10% heat-inactivated fetal calf serum (FCS) and grown at 10% $CO_2$ at 37 °C. Cell lines were split 1:4 every 3 days using trypsin. 293 reporter cell lines harboring a lentivector expressing destabilized GFP under the control of IFN-stimulated responsive elements (ISRE) or destabilized GFP under control of an IFN-β promoter were a kind gift from Jan Rehwinkel[35,36]. Note that these reporter constructs include a puromycin cassette, hence we did not use puromycin-selection of shRNAs in these cells, which was unnecessary due to their high transduction efficiency. Human primary foreskin fibroblasts (HFFs) were a kind gift from Matt Reeves and early-passage cells were split 1:3. shRNAs were puromycin-selected in HFFs. The THP-1 IFIT-2 *luciferase* reporter cell lines were purchased from Invivogen and cultured according to the manufacturer's guidelines. These cells are WT or CRISPR/Cas9 knockout for cGAS, STING or IFNAR and express secreted Lucia *luciferase* under control of a minimal IFIT2 promoter in conjunction with 5 ISRE units. IFN-β (PeproTech) or Universal type 1 IFN (Interferon-αA/D, PBL Assay Science) were used at 10 ng/ml or 200 U/ml, respectively. Poly (I:C) (Invivogen) was used at 150 ng per well in a 24 well plate and dsRNA was made by in vitro transcription[61] and used at 1 μg/ml complexed with Lipofectamine 2000 (Invitrogen, Thermo Fisher Scientific) at a 1:3 ratio according to the manufacturer's instructions. The reverse transcriptase inhibitors, AZT and 3TC were used at 100 μM and 200 μM, respectively. The JAK-STAT inhibitor Ruxolitinib (Cambridge Bioscience) was used at 2 μM, which was determined by titration on ISRE-GFP treated with IFN-β.

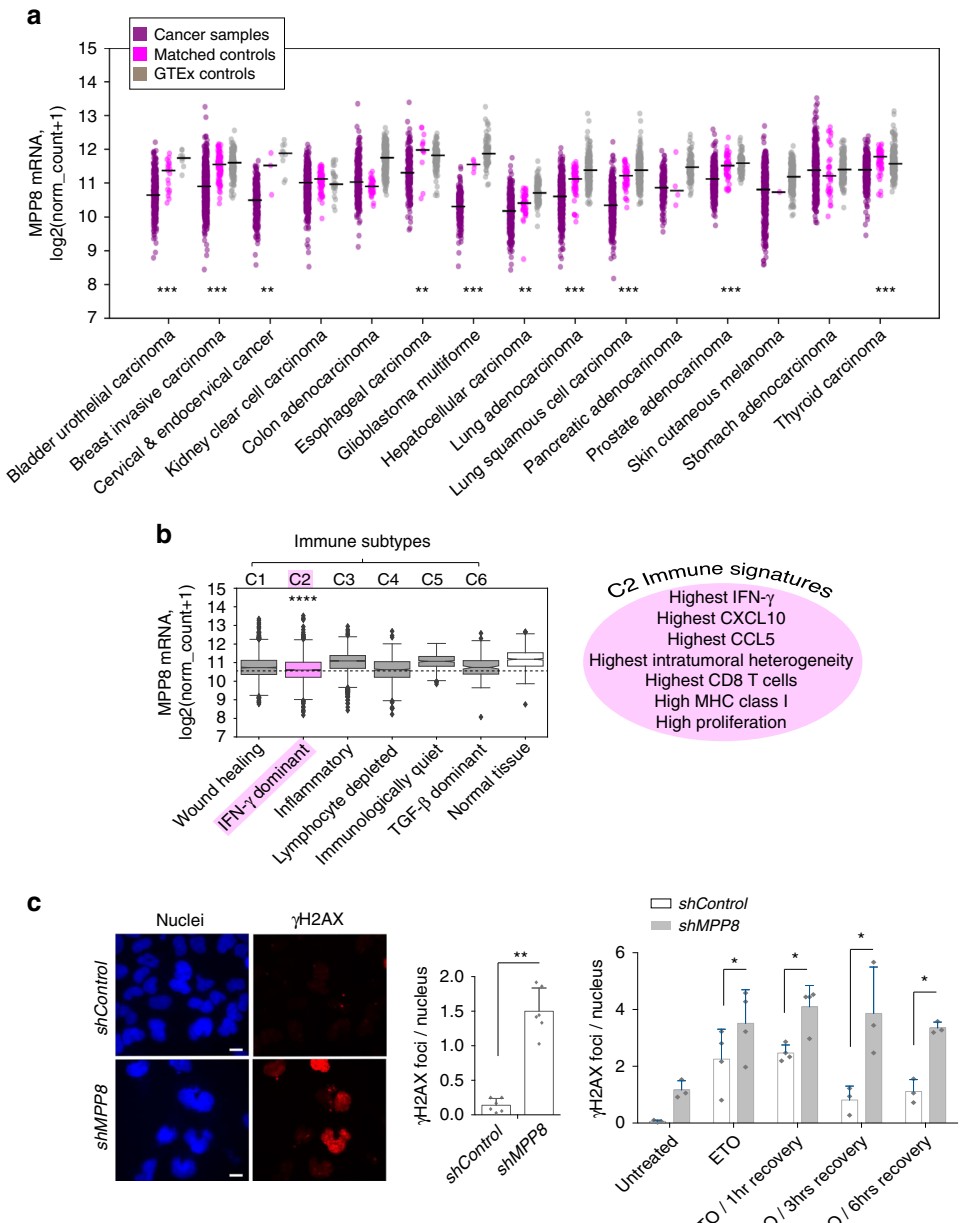

**Fig. 6 MPP8 is frequently downregulated in cancers. a** Using The Cancer Genome Atlas (TCGA) data, we selected cancers for which there were matched control samples as well as GTEx expression data from the same tissue. MPP8 is scored here as significantly downregulated when levels are lower in the cancer compared to both controls separately, using a two-sided Mann–Whitney rank test and correcting for multiple testing (*p* adjusted values were all <0.05 for groups marked with asterisks). Adjusted *p* values for matched control comparisons and GTEx comparisons respectively are given here: bladder urothelial = 2.40e−05, 1.58e−06; breast invasive = 5.40e−35, 8.79e−61; cervical & endocervica l= 3.12e−03, 2.37e−07; esophageal = 1.07e−03, 4.0e −15; glioblastoma = 2.10e−08, 6.61e−40; liver hepatocellular = 1.31e−03, 5.57e−31; lung adeno = 1.19e−11, 9.56e−82; lung squamous cell = 2.40e−26, 1.41e−97; prostate adeno = 2.06e−11, 8.27e−31; thyroid = 1.25e−10, 9.24e−17. **b** Left: all cancers were divided by immune subtype based on ref. [17] and levels of MPP8 expression were plotted. MPP8 levels were most significantly different in subtype C2 compared to normal tissue (after pairwise, two-sided *t* tests between each immune subtype and normal tissue with an fdr multiple test correction: padj = 8.03e−94; C1 *n* = 2,066, C2 *n* = 2413, C3 *n* = 2334, C4 *n* = 1129, C5 *n* = 378, C6 *n* = 180 samples). Boxes represent 1st and 3rd quartile and the central line is the median. Notches indicate the confidence interval; whiskers correspond to 1.5 of the interquartile range and outliers are depicted as diamonds. Right: features of cancers within immune subtype C2 are displayed (terms from ref. [17]). **c** 293 reporter cells were treated with *shRNAs* and then either untreated or treated with etoposide (ETO) for 30 min and allowed to recover for stated times before γH2AX staining. Left: example γH2AX staining images. Middle: summary data of γH2AX foci: *N* = 6 biologically independent samples. *p* = 0.0013 (two-tailed paired *t* test). Right: summary data of recovery assay. *N* = 4 biologically independent samples, except for the groups, ETO and ETO/1 h, where *N* = 3. Two-tailed paired *t* test *p* values = 0.0303 (ETO) 0.0486 (ETO/1 h) 0.0384 (ETO/3 h) and 0.0258 (ETO/6 h). Data show the mean ± SD. See Supplementary Fig. 6e for HeLa data.

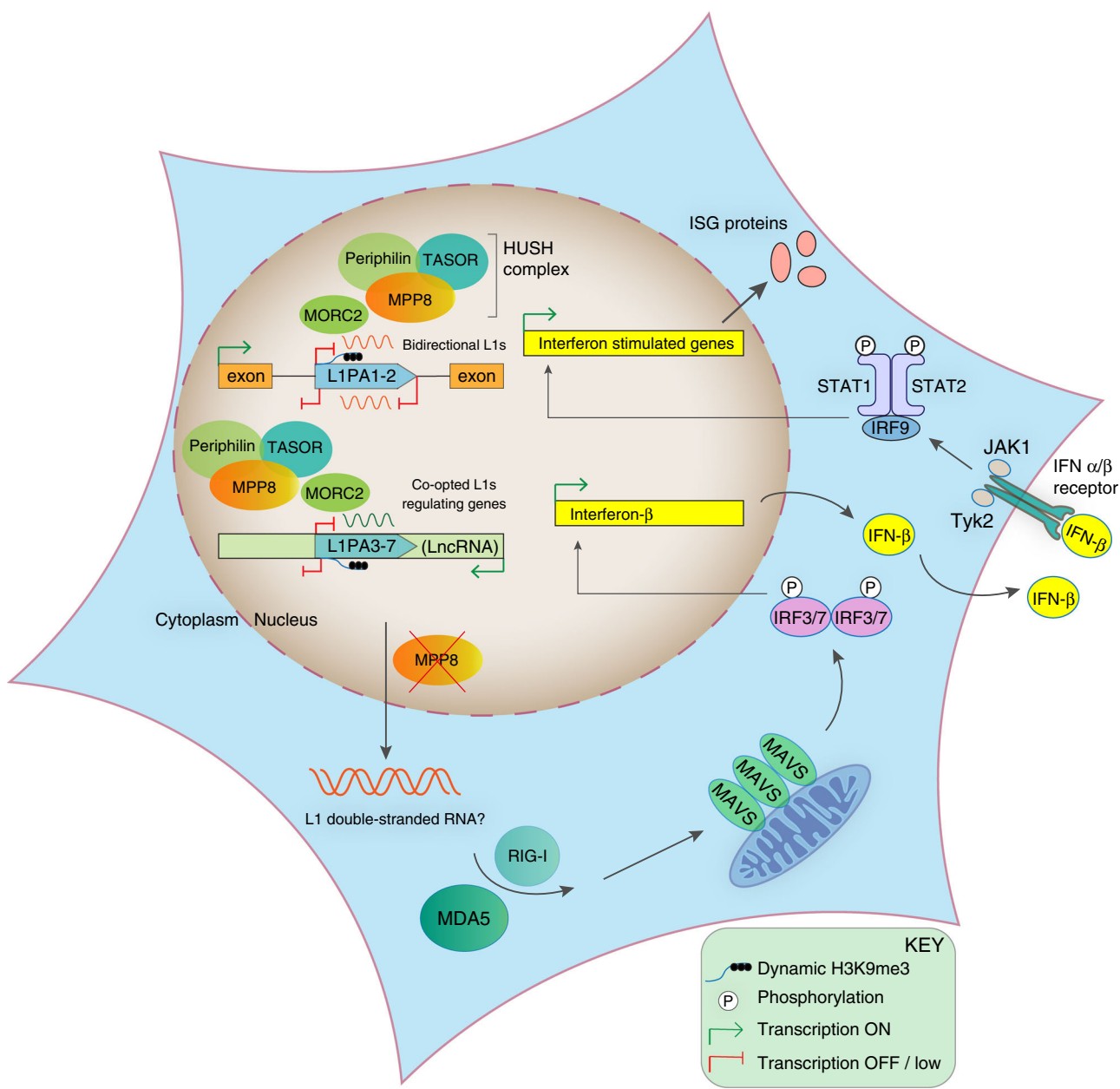

**Fig. 7 Proposed model.** Inactivation of the HUSH component MPP8 leads to increased expression of bidirectionally-transcribed full-length young LINE-1s (L1PA1 and L1PA2), as well as primate-conserved LINE-1s with potential co-opted roles in gene regulation, (some enriched in long non-coding RNAs). Nucleic acid sensing ensues through dsRNA sensors MDA5 and RIG-I, leading to MAVS signaling and activation and secretion of type I IFNs. IFN-α/β then act on the same cells or bystander cells to mediate induction of IFN-stimulated genes through JAK-STAT signaling. MPP8-depletion is also accompanied by DNA damage and is a factor commonly downregulated in cancers, where its decreased mRNA expression is associated with a dominant IFN immune subtype signature.

**Western blotting**. Cells were washed once in ice-cold PBS and lysed using pre-chilled homemade Radioimmunoprecipitation assay buffer [150 mM NaCl; 1% Triton X-100; 0.5% sodium deoxycholate; 0.1% SDS and 50 mM Tris, pH 8.0, protease inhibitors (#539134, Calbiochem) and phosphatase inhibitor cocktail (PhosSTOP, Roche)] for 30 min at 4 °C. Cell lysates were cleared by centrifugation (12,000 × g, 20 m min, 4 °C) and a fraction of lysate was used to perform a protein quantification assay (BCA Protein Assay Kit, Novagen). Standardized lysate samples were mixed with NuPAGE LDS sample buffer (Thermo Fisher) and 5% β-mercaptoethanol and heated at 95 °C for 5 min and then resolved on precast NuPAGE 4–12% Bis-Tris (Invitrogen) or handcast 10% SDS-polyacrylamide gels in tris/glycine/SDS buffer in mini-PROTEAN tanks (Biorad), followed by wet transfers onto Polyvinylidene Difluoride (PVDF) or nitrocellulose membranes, blocked in 5% non-fat dried milk in TBS-T (TBS, 0.1% Tween-20) and incubated with relevant primary antibodies: anti-MPP8 (#16796-1-AP, Proteintech), anti-TASOR/FAM208A (HPA017142, Atlas Antibodies), anti-PPHLN1 (HPA038902,

Atlas Antibodies), anti-L1 ORF1p (MABC1152, Millipore), anti-PCNA (NA03, Calbiochem), anti-α-Tubulin (T6074, Sigma Aldrich), anti-RIG-I (D14G6, #3743, Cell Signaling Technology), anti-MDA5 (D74E4, #5321, Cell Signaling Technology), anti-MAVS (#3993, Cell Signaling Technology), anti-STING (D2P2F, #13647, Cell Signaling Technology), anti-cGAS (D1D3G, #15102, Cell Signaling Technology), anti-IFIT1 (D2X9Z, #14769, Cell Signaling Technology) and anti-ISG15 (A-5, sc-166712, Santa Cruz Biotechnology). All secondary antibodies were horseradish peroxidase-conjugated (GE healthcare) and membranes were developed using ECL kits (ECL, Prime or Select kits from Amersham). Source Data are available with original blots.

**shRNA lentiviral vectors and transduction**. Hairpin sequences were designed against LINE-1 integrants (using the 5′UTR or ORF1 or ORF2 sequence) known to be bound by MPP8[33] (see Supplementary Table 1) and using the Clonetech RNAi

designer website (http://bioinfo.clontech.com/rnaidesigner/) and verified in silico before cloning. Hairpin primers were annealed and then cloned into an shRNA vector[15] at *BamHI-EcoRI* sites and the products were checked by verification of an introduced restriction site and by sequencing. VSV-G-pseudotyped lentiviral vectors were produced by co-transfecting 293T cells in 10 cm plates with 1.5 μg of the shRNA plasmid, 1 μg p8.91 and 1 μg pMDG2 encoding VSV-G. Media was changed 1 day post transfection and supernatant harvested 48 h post-transfection and concentrated via ultracentrifugation. Puromycin selection was performed 3 days post transduction overnight or until control cells died. For 293 and HeLa cells, puromycin selection was not necessary and for this reason, dual transductions were performed in 293 cells.

**CRISPR/Cas9 generation of RIG-I-like receptor knockout HEK293 cell lines**. RIG-I, MDA5, and RIG-I/MDA5 knockout 293 cells were generated by CRISPR–Cas9-mediated genome engineering. Using the CRISPR design tool provided by the Zhang lab (www.genome-engineering.org) several guide sequences were selected in human *DHX58* (encoding RIG-I) and *IFIH1* (encoding MDA5) (Supplementary Table 1). Appropriate oligonucleotides corresponding to these guide sequences were cloned into the BbsI site of pX459 (Addgene 62988), a bicistronic expression vector encoding both Cas9 and the single-guide RNA, according to the cloning protocol provided by the aforementioned website. Plasmid pX459 was obtained from the lab of Dr. Feng Zhang via Addgene (plasmid 62988)[62]. 293 cells were transfected in a 6-well plate using 1 μg pX459 per well using Lipofectamine 2000 (Life Technologies) according to the manufacturer's instructions. Twenty-four hours post-transfection, cells were treated for 24 h with 1 μg/ml puromycin to remove untransfected cells and subsequently replated at limiting dilution in order to pick individual colonies. Knockout of RIG-I and/or MDA5 was verified by amplification and sequencing of the genomic region surrounding the CRISPR target site, immunoblotting, and in a *luciferase*-based transfection reporter assay, in which *luciferase* activity is controlled by the IFN-β promoter, using selective RIG-I and MDA5 agonists.

**RNA extraction and quantification**. Total RNA was extracted using RNeasy Micro kit columns (Qiagen) and DNase treated according to the manufacturer's instructions (Ambion AM1907). 500 ng of RNA was reverse transcribed using random primers and SuperScript II Reverse Transcriptase (Thermo Fisher Scientific). Control reactions were always performed in the absence of reverse transcriptase and used for qRT-PCR in parallel to cDNA to verify there was no DNA remaining. cDNA was diluted in nuclease-free water, and gene expression levels were quantified using quantitative reverse transcription PCR (qRT-PCR) using an ABI 7500 Real Time PCR System (Applied Biosystems). SYBR green Fast PCR mastermix (Life Technologies) was used. CT values for the test genes were normalized against those of *Gapdh* or *B2M* using the −ΔΔCt method to calculate fold change. See Supplementary Table 2 for primer sequences, which includes a breakdown of TE primer hits to TE subfamilies.

**Flow cytometry**. Cells were trypsinized and harvested in media and centrifuged and then fixed in 1% paraformaldehyde for 10 min (or unfixed) and washed in PBS, resuspended in PBS and run on a BD FACSCalibur or LSRFortessa Flow Cytometer acquiring 10 000 cells per sample to meet statistical robustness using BD CellQuest or FACSDiva software. Intracellular FACS[34] was performed in 293 cells (not reporter cells) using the dsRNA-specific antibodies, J2 and K1 (from English and Scientific Consulting Hungary, SCICONS) or an isotype control antibody (mouse IgG2a, Santa cruz sc-3878), with a secondary Alexa Fluor 488 goat anti-mouse IgG (ThermoFisher). We verified the antibodies to recognize dsRNA by intracellular FACS after transfection of dsRNA. ShortCut RNase III was used for long dsRNA digestion (NEB). Data were analyzed using FlowJo (Tree Star version 10.3.0). The flow cytometry gating strategy, which involved gating on live cells and then on GFP positive cells (or dsRNA-positive cells) using the negative control sample is shown in Supplementary Fig. 7.

**Luciferase assays**. 293T cells were plated at $10^5$ cells/ml in 12 well plates, 1 ml per well and co-transfected with 500 ng KZNFs, 50 ng SVA or L1PA4 *luciferase* reporter plasmids (kind gift from David Haussler[40]) and 5 ng pRT-TK *Renilla* control plasmid. Cells were then lysed and *luciferase* measured 48 h later using the Dual Luciferase assay kit (Promega, E1910) and a Glomax 96 microplate Luminometer (Promega) using the Dual Glow Programme according to the manufacturer's guidelines.

**Interferon-β ELISA**. An ELISA kit precoated with an antibody against human IFN-β (VeriKine Human IFN Beta ELISA Kit, 41410 by PBL) was used according to the manufacturers' instructions. Absorbance measurements were taken at 450 nm using a Multiskan FC Microplate Photometer and SkanIT Software. Optical densities were plotted using a 4-parameter fit for the standard curve and this was used to determine the IFN-β concentrations of test samples.

**Engineered LINE-1 reporter constructs**. RNA-sequencing data were employed from ref. [21], in which a Tet-inducible codon-optimised L1PA1 (ORFeus) was induced in RPE cells using doxycycline treatment for 3 days. vs. an induced *luciferase* control. A list of genes that were significantly upregulated (>10 fold with *p* adjusted values <0.05) was obtained (for data access, see: https://www.ncbi.nlm.nih.gov/geo/query/acc.cgi?acc=GSE119999) and the overlap of this list compared to genes upregulated in this study (HFF cells) upon depletion of MPP8. For experiments in which we overexpressed engineered LINE-1, we used a full-length active L1PA1 construct expressed from a CMV promoter (ksCMV-101/L1.3, which is not codon-optimized, kind gift from Jose Garcia-Perez). This construct contains a neomycin resistance cassette, inserted within the 3′UTR, in the opposite direction, interrupted by an intron. Expression of the neomycin gene depends on transcription, splicing, reverse transcription and integration. Engineered L1PA1 was transfecting into 293 ISRE-GFP reporter cells vs. pcDNA control and 4 days later, L1 ORF1 protein, spliced Neomycin expression and ISRE-GFP reporter induction were measured.

**Total RNA-sequencing and analysis**. The RNA was quality checked on a 4200 Tapestation using the RNA ScreenTape assay (Agilent Technologies, Wokingham, UK) and the RNA concentration measured using a Qubit RNA Broad Range kit (Life Technologies, Paisley, UK). Total RNA samples were then processed using KAPA's stranded RNA HyperPrep RiboErase kit using an input of 500 ng per sample. Samples were sequenced on a NextSeq 500 instrument (Illumina Cambridge, Chesterford, UK) after pooling libraries in equimolar quantities, using a 2x 151 bp paired-end run, resulting in over 15 million reads per sample. Illumina's bcl2fastq Conversion Software was used to demultiplex data and generate fastq files. Fastq files were checked for quality with FastQC v.0.11.8[63] and TrimGalore v0.4.1[64] was used to remove adaptors only. Reads trimmed for adaptors were aligned to the human genome USCS build hg38 using Tophat 2.1.0[65] and Gencode v30[66] gene annotations (https://www.gencodegenes.org/human/release_30.html). The number of read counts per gene was calculated using HTSeq-Count[67], the BioConductor package DESeq2[68] was used to estimate differential expression of genes across conditions, and the Approximate Posterior Estimation method[69] was used to shrink the logarithmic fold change. As per the default behavior of DESeq2, p-values were adjusted for multiple testing with the Benjamini–Hochberg false-discovery-rate (FDR) procedure. Genes were considered as significantly differentially expressed when the adjusted *p*-values were <0.05, and where the log2 of the fold change was >1 for upregulated or < (−1) for downregulated genes. BiomaRt[70] was used to convert gene identifiers and gene set enrichment analysis, based on rank derived from the $abs(\log2FC)*(-\log10(pvalue))$ and the Hallmark Gene Set Collection from MSigDB[71,72] (https://www.gsea-msigdb.org/gsea/msigdb/genesets.jsp?collection=H), was performed with the fgsea package 1.8.0[73] and gage 2.32.0 from R Bioconductor. We defined the list of ISGs before[15] by employing http://www.interferome.org and selecting genes upregulated by 10-fold upon IFN treatment. Enrichment of Reactome pathways (https://reactome.org/) within gene subsets was performed with clusterProfiler[74]. For analysis of individual TE integrants, TopHat2 alignments were used to obtain the number of unique reads mapping to individual TE loci using HTSeq-count and custom-made annotation files where each TE locus was given a unique identifier. To reduce the number of features provided as input for the analysis, TE loci with less than 20 counts across all 21 samples were filtered out before submitting the files with the counts per TE loci to DESeq2. Differential expression analysis was performed as for genes and including gene counts. Venn diagrams were made with VennDiagram 1.6.20.

For analysis of TE family expression, the TEcounts tool of TEtranscripts[39] was used to obtain TE counts, followed by differential expression analysis using DESeq2. TE families were considered to be significantly differentially expressed when the adjusted *p*-values were <0.05, and where the log2 of the fold change was >1 for upregulated families and < −1 for downregulated families. HUSH binding to TEs was assessed using public data[33] found here: https://www.ncbi.nlm.nih.gov/geo/query/acc.cgi?acc=GSE95374.

Expression coverage tracks were generated with the *genomecov* tool from bedtools[75], scaled by library size, where read pairs were separated by their strand of origin and individual replicates were merged for knockdown and control samples of biological triplicates from the same batch. Different vector preparations (produced at different times) were used for each biological replicate. bigWig tracks were visualized using the Integrative Genomics Viewer[76]. For identification of bidirectional transcription, genome-wide coverage was calculated for 500 bp bins, using the above-mentioned BigWig tracks. A bin was considered to be expressed from a given strand if its mean expression was above the value equivalent to the 85th percentile of all bins with an expression value greater than 0. Bins that passed the threshold for both strands were considered to be bidirectionally transcribed. Bidirectionally transcribed bins were merged into larger intervals when found adjacent to each other using bedtools merge. We then intersected results with RepeatMasker annotation (http://www.repeatmasker.org/) to uncover bidirectional TEs, with the top hit being L1PA1.

For the TEs with the largest number of loci within bidirectionally-transcribed bins (L1PA1 and L1PA2), we retrieved consensus sequences from RepBase (https://www.girinst.org/repbase/) and aligned all reads to each individual reference (full-length L1PA1, full-length L1PA2, assembled from RepBase fragments: L1P1_5end, L1P1_orf2, L1PA2_3end, or the sequence of a full-length L1PA2 previously identified to be MPP8-bound[33], coordinates: chr1: 207733982-207734498). Depth across the different sequences was calculated using samtools depth, normalized

using size factors calculated for the entire library. Total number of reads mapping to a reference was obtained with the samtools flagstat command.

Each TE family in the human genome was assigned a taxonomy level using information from Dfam (https://dfam.org/). The relative abundance per taxonomy level of all families in the genome or upregulated families only was visualized in a phylogenetic tree obtained from the NCBI Taxonomy Database (https://www.ncbi.nlm.nih.gov/ Taxonomy/Browser/wwwtax.cgi) using the ETE Toolkit[77].

The distance between the TE subsets (L1s or LTRs upregulated in shMPP8 samples compared to shControl samples) and the transcription start sites (TSSs) of IFN response genes was calculated using the pybedtools closest function with RepeatMasker annotation and TSSs retrieved using biomaRt (https://www.ensembl.org/biomart/martview/). Random subsets of genes were generated within python. Total RNA-sequencing data can be found here: https://www.ncbi.nlm.nih.gov/geo/query/acc.cgi?acc=GSE135765.

**The cancer genome atlas (TCGA) and GTEx analysis**. Phenotype data were downloaded from https://xenabrowser.net/ for the TCGA TARGET GTex cohort[78]. The phenotype table was manually curated to include only cancer types that had both a matched normal tissue from TCGA and an independent normal tissue from GTex. The *TcgaTargetGtex_RSEM_Hugo_norm_count* dataset was queried using the xena-Python python API. Differences in expression between cancer and control samples were statistically assessed using a Mann–Whitney *U* test followed by an FDR multiple test correction in python. The results shown in Fig. 6 are in part based upon data generated by the TCGA research network (https://www.cancer.gov/tcga)[44], The Genotype-Tissue Expression (GTEx) Project[45] and The TARGET Program (https://ocg.cancer.gov/programs/target)[78]. The Genotype-Tissue Expression (GTEx) Project was supported by the Common Fund of the Office of the Director of the National Institutes of Health, and by NCI, NHGRI, NHLBI, NIDA, NIMH, and NINDS; the data used for the analyses described in this manuscript were obtained from dbGaP accession number phs000424.vN.pN on 01/09/2019. Raw data from Synapse ICGC-TCGA Whole Genome Pan-Cancer Analysis project (https://www.synapse.org/#!Synapse:syn2351328/wiki/62351, https://www.synapse.org/#!Synapse:syn11288411) was downloaded and visualised using UCSC Xena (https://doi.org/10.1101/326470)[78]. Immune subtype data were retrieved from https://xenabrowser.net/datapages/?dataset=Subtype_Immune_Model_Based.txt&host=https%3A%2F%2Fpancanatlas.xenahubs.net&addHub=https%3A%2F%2Fxena.treehouse.gi.ucsc.edu&removeHub=https%3A%2F%2Fpcawg.xenahubs.net. Source data are available with RNA-seq datasets.

**DNA damage**. HEK293 or HeLa cells were transduced with shRNAs and then $2 \times 10^4$ cells plated per well two days later in 96 well Cell-carrier optical plates. 4–6 days post transduction, cells were treated with 60 μM or 30 μM etoposide for 30 min or left untreated. For recovery studies, etoposide-treated cells were washed 2× in PBS and new media was added for the indicated times. Cells were fixed with 3% paraformaldehyde, quenched with 50 mM $NH_4Cl$ and permeabilised with 0.3% Triton X-100 in PBS. Blocking was performed in PBS containing 1% FCS before washing and staining with the γH2AX primary antibody for 1 hour and then washed and stained for 1 h with a Red Alexa Fluor-conjugated secondary antibody. Cells were then washed and stained with DAPI. Images were acquired and analyzed using a Hermes WiScan-automated system (IDEA Bio-Medical Ltd. Rehovot, Israel), using ImageJ software.

**Statistical analysis**. All data are presented with error bars showing standard deviation (SD) or standard error of the mean (SEM), where stated and statistical significance was assessed using two tailed, unpaired Student *t* tests, or other statistical tests where stated (see figure legends for details) employing GraphPad prism. The number of biological replicates is stated in the figure legends and technical replicates where relevant: for flow cytometry technical replicates were 10,000 events. A *P*-value of <0.05 was considered statistically significant (****$p < 0.0001$, ***$p < 0.001$, **$p < 0.01$ and *$p < 0.05$).

**Reporting summary**. Further information on research design is available in the Nature Research Reporting Summary linked to this article.

## Data availability
Total RNA-sequencing data are available on the NCBI Gene Expression Omnibus database (accession number, GSE135765) and other data are included in this article and its supplementary information files. Accession numbers for the publicly-available data are: GSE95374 for ChIP-sequencing data on the HUSH complex, and GSE119999 for RNA-sequencing data on cells expressing engineered LINE-1s. All data is available from the authors upon reasonable request. Source data are provided with this paper.

## Code availability
Code is deposited in the following open source repository (https://github.com/regmdr/HUSH_analysis) and detailed in the methods and all code can be retrieved by contacting the corresponding author.

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

## Acknowledgements

We thank Jose Garcia-Perez for the engineered L1PA1 construct, Jan Rehwinkel for the ISRE & IFN-β reporter HEK293 cell lines, Matthew Reeves for the primary fibroblasts and Paul Lehner and Caetano Reis e Sousa for advice. We thank the Pathogen Genomics Unit at UCL run by Judy Breuer for total RNA-sequencing runs, and the TCGA and GTEx consortiums for public expression data[44,45]. This work was funded through a Sir Henry Dale Fellowship through the Wellcome Trust and Royal Society (Grant number 101200/Z/13/Z, supporting H.M.R., H.T., C.H.C.T., and J.H.) and a European Research Council starting grant (678350, TransposonsReprogram, supporting R.E-G., P.G., L.F., and H.M.R.). P.M. and R.G. are funded by a Wellcome Trust Senior Fellowship to R.G. (WT108082AIA). P.V.M. received funding through a Wolfson UCL Excellence Fellowship and is funded through a UKRI Future Leaders Fellowship (MR/S034498/1). This work was also supported through a consumables grant funded by the Rosetrees Trust and Robert Luff Foundation (A2630) and a Barts Charity Lectureship (MMBG1R) awarded to H.M.R. This work is dedicated to the memory of Paul Rowe.

## Author contributions

H.T., R.E-G., C.H.C.T., and H.M.R. conceived, designed, and performed experiments, analyzed data and wrote the paper. P.G., P.M., R.G., L.F., J.H., A.G.V., E.G., K.H.B., and P.V.M. performed experiments and contributed to ideas. All authors read and approved the final paper.

## Competing interests

The authors declare no competing interests.
