## [Peer Review File · Nature Communications]

Reviewers' comments:

Reviewer #1 (Remarks to the Author):

In "The HUSH complex is a gatekeeper of type I interferon induction through epigenetic control of LINE-1s", the authors posit that LINE1 activity induces a type-1 interferon response via their RNA, which is detected by MDA5, propagated through MAVS signaling and executed by JAK-STAT signaling. They also provide evidence that the absence of the HUSH complex actively suppresses LINE1s to prevent this specific pathway of interferon induction, which is primarily evidenced using knockdown of MPP8 with the various assays presented. They conclude this report by providing evidence that the many human cancers elicit a downregulation of MPP8 and that this may contribute to the elevated DNA damage observed in these cancer cell lines.

In general, the authors have provided an interesting case for LINE1 regulation and a novel pathway by which LINE1 activity results in induction of inflammation. Previous reports by (De Cecco, et al, 2019 Nature) and (Simon, et al, 2019 Cell Metabolism) have indicated that LINE1 cytoplasmic DNA and sensing by the cGAS/STING pathway also induce inflammation, but the work done here by Tunbak, et al. proposes a novel pathway in which the formation of dsRNA triggers a similar response. With the additional link to the HUSH complex as a chief regulator of LINE1 activity, this work outlines an exciting new mechanism by which LINE1s are regulated and can produce a pathological response.

While the evidence presented is suggestive of the conclusions the authors have drawn, there are several major concerns that need to be addressed prior to publication of this manuscript. Overall, the manuscript needs many additional replicates and appropriate presentation of the data to be properly assessed. Until the lack of rigor is addressed, I cannot recommend publication. That said, if the authors can address these issues, these findings are of considerable interest to the field of transposon biology and its role in disease.

Major Issues

- 1.) In many cases, the data presented lacks an adequate amount of rigor. A great deal of the data presented is denoted as "one representative experiment of two shown". At the bare minimum, at least 3 replicates of an assay need to be conducted and the data should be presented as an average with the error indicated. Additionally, any representative figures should be accompanied by quantification from the 3 separate trials. This misstep is prevalent throughout the manuscript and must be rectified.
- 2.) Accompanying the previous point, all data should have error bars where appropriate.
- 3.) The authors combine data from multiple time points in 1C, 1F, and 3B. This is not appropriate and they should have replicates for a single time point.
- 4.) The presentation of qRT-PCR is inappropriate. Each target should be presented with its own error bars. The combining of several targets and attempting to interpret statistics off of the clustering is not a valid approach. It is not clear how many technical replicates were used per target.
- 5.) A major caveat of this work is the almost explicit use of HEK393 cells (and the occasional Hela). This makes it difficult to assess the relevance to normal cellular biology, as both the cell lines are far removed in behavior and epigenetics from primary human cells. The addition of a primary cell line to some of these experiments would greatly benefit the impact of these findings
- 6.) While the authors used NRTIs and demonstrated a lack of effect on the inflammation response, it has been reported that LINE1 activation stimulates both interferon α and β (De Cecco, et al. 2019). I do not necessarily doubt the role of dsRNA in the inflammation response the authors observed, but they should address this discrepancy. Analysis of cGAS/STING via western/qRT-PCR would be prudent. Additionally, the authors should provide evidence of interaction of LINE1 dsRNA with MDA-5 (immunostaining or FISH would be appropriate).
- 7.) Related to the above point (#6), this discrepancy with earlier publications may be the result of using highly mutated cancer cell lines in this work, which may lack functional cGAS/STING pathway. Repeating the experiments in normal cells is therefore essential. Otherwise the

conclusion should be toned down to apply to cancer cell lines only.

8.) The authors should provide data for PPHLN1 knockdown for Figures 1F-H in order to concretely claim the involvement of the HUSH complex. The reliance on MPP8 for the subsequent work would be permissible if this was provided.

Additional Points

- The manuscript needs to be edited for clarity. There are many grammatical errors.
- The authors should include more about the role and function of the HUSH complex in the introduction. Additionally, the logic in Figure 2 should be expanded upon. The authors appear to focus on the ZNFs without addressing the other 23 non-ISG genes
- In general, the authors have enough space for figures that they should not have multiple graphs under the same subfigure (See Figure 1F, 1G, 3C, 3D, 4F, Figure 5)
- Initial RNAi experiments should demonstrate the effects using at least 2 separate shRNAs to the same target to assuage off target contributions
- Figure 1F appears to have mismatch of colors in the graph vs the legend. Also, why aren't the other ISGs assayed in the HeLa cells
- Figure 3A appears to be missing the shControl + shMAVS for the ISRE-GFP set
- Figure 3C lower error bar missing on first two samples
- Figure 3D western for Rig-1 is not publishable. Please get a clearer image.
- Figure 3D expression is not elaborated on in Figure legend. Is this qRT-PCR data or quantification of the western?
- Figure 4A is missing a box
- Figure 4E needs to be quantified. Additionally, there needs to be an explanation for the discrepancy in the ORF1p size between the HEK293 and HeLa cells (one is ~50kd and the other is ~37kd)
- Figure 4F qRT-PCR targets need to be separated. Combining different targets for analysis is inappropriate
- Figure 4G. Wouldn't the authors expect an overlap between these two data sets to be larger? There should be some quantification and statistical statement on whether this level of overlap is even relevant. Additionally, the unpublished data should be included in this manuscript if it is to be used.
- yH2AX foci are typically quantified. Figure 6B should be averaged counts from cells, not an intensity plot. Intensity is not reliable way of quantifying DNA damage for yH2AX

Reviewer #2 (Remarks to the Author):

In the manuscript entitled " The HUSH complex is a gatekeeper of type I interferon induction through epigenetic control of LINE-1s", Tunbak and colleagues describe a group of Transposable Element/s linked to the induction of a type I INF response, which is a very relevant question in human biology.

While some parts of the study are solid, the manuscript lacks key controls and some results are OVERinterpreted, decreasing the robustness and accuracy of the study. The cancer connection is very weak, and adds nothing to the study. Additionally, some data shown in Figures don't support text claim's, which is quite confusing. Finally, the term "young L1" is arbitrarily and inaccurately used by authors, making the study confusing and wrong. For example, authors refer to elements regulated by ZNF93 (L1PA3 and older, that is older than 12 Millions of years) as young elements, which is inaccurate.

With these considerations, I would not recommend publication of this study in its current form, unless much more definitive evidence is provided regarding the causative role of young L1 RNAs in generating a MPP8-dependent INF response.

Major points

1) The role of TASOR. Authors suggest that TASOR depletion could rescue the effect mediated by shRNAs to periphilin, and that this could occur through activation of a negative immune regulator. Is there any evidence for this? Can author check this?

Also, is the same observed in other cellular backgrounds? As done in Figure 1F, what's the effect of TASOR shRNAs on other cell lines?

2) Can authors demonstrate that the HUSH complex regulates type I INF induction on a non-transformed cell line? This is important, as karyotypic instability increase the overall copy number of TEs per cell, and what authors are reporting could be completely artefactual. At the very least, some proof of principle experiments on HCT116 (nearly diploid) should be provided.

3) Experiments using JAK-SAT inhibitors lack proper controls, and authors should show data regarding the effectiveness of the treatment by exploring a known target.

4) How deep is the phenotype scored in Figure 2? As the supernatant from MPP8 depleted cells is sufficient to activate an INF response (Fig 1), data shown in Fig2 correspond to a mix of cells with an activated INF response and cells responding to secreted ILs etc....thus, unless these experiments are done in a single cell manner, as presented the data is misleading, and the contribution of MPP8 could be quite small. It would be critical to conduct these assays in a karyotypically stable cell line, as much of the misregulation could be driven by increased copy number of transposons and retrotransposons.

5) Authors used HeLa cells to identify genes differentially regulated upon MMP8 depletion (Fig 2). Then, the RNAseq data is used to identify potentially misregulated TE RNAs, as their data suggest that the ligand driving the INF response is endogenous. As HeLa cells are known to harbour hypermethylated L1 promoters, and as their RNAseq dataset correspond to early time points, it is not surprising that LINE-1 RNAs are not much upregulated, Thus, unless additional time points are analyses, these analyses are incomplete. Indeed, and as stated above, these analyses should be ideally done in a cell with a normal karyotype, and in a single cell manner, to ensure that what authors are reporting is not linked to overall increased number of LINE and SINE elements and/or in response to secreted ILs.

6) Figure 4B, which is important to support the model proposed by authors (that L1 RNAs are rapidly suppressed by ZFPs), need to be expanded. As it is, there is only a very childish scheme. Although some data is shown in Figure 2D, authors need to provide compelling evidence that indeed these ZFPs can bind L1, and more importantly describe which LINE-1 subfamily is deregulated. There are many LINE subfamilies in the human genome, and this information is critical. Related to above: RNAseq analyses to identify depressed TEs should be performed using Tetrascript and/or SQUIRE, rather than an internal pipeline (there is not enough information to judge the quality of the bioinformatic approach used by authors). This is critical, as the claims are important and likely to be reproduced by others.

7) Figure 4C: completely non informative. As above, there are >100 LINE and SINE subfamilies annotated in the human genome. Thus, specific subfamily names should be included here. This apply to LINES, SINES, ERVs, DNATpson and Bidirectionally transcribed TEs.

8) Authors NEVER demonstrate that indeed 5kb dsRNAs might be produced, as claimed in page 9 lanes 189-190. Thus, evidence for this should be provided. This is critical to support the model proposed by authors, which indicate that the INF response is mostly dependent on the double-stranded RNA sensor MDA5. RNA Protection Assays would be ideal for this.

9) Figure 4D. These data is very problematic. On the left caption, it's quite clear that the shControl and shMPP8 generate a similar increase in transcription, from the same genomic point. Why authors state that: "Most of these instances involved LINE1 elements within genes whereby LINE1 RNA was produced in the opposite sense to the genic transcript, illustrating how double-stranded RNA of around 5kb might be produced (examples are shown in Figure 4D..." Very confusing. The example shown in the right side is consistent with the text claims, but there is no way to guess the length of the potential dsRNA generated.

10) Figure 4E. It's critical to demonstrate that young L1s are derepressed in response to MPP8 depletion, as the antibody used is directed to the younger L1Hs subfamily of L1, and it's a monoclonal antibody. The data provided in Fig4C is non-informative and the two examples shown

in Fig 4D (beside been problematic) correspond to evolutionary older L1s that would not be recognized by the antibody used (L1PA7 and L1M1, 30 and 61 Millions of years).

11) Figure 4E. Don't support the claims of the authors. RNAseq analyses were done at day 3, and it's quite clear that there is no L1-ORF1p overexpression at day3. This might not be surprising, as there is a reported delay L1 in translation (see Kulpa and Moran, Hum Mol Genet 2005). Thus, a Northern blot and/or RT-qPCR should be used to really confirm that MPP8 depletion result in L1 RNA overexpression at day 3, presumably from the L1Hs subfamily according to the western data shown in Figure 4E.

12) The experiments with TSA are OVERinterpreted, and only provide indirect evidence that histone acetylation might control L1 expression (already published in reference 29). The experiment is poorly described (cell type?, concentration?, time?), but most importantly lacks control. How surprising is that TSA, a very strong IHDAC, can change the transcriptome of cells? This has been reported before, and again, there are no proper controls in this experiment. Consistently, the overlap with the Ardeljan study is minimal (16 of 77 markers). Genetic evidence should be provided here. More importantly, young L1s, which are presumably deregulated in response to MPP8 depletion according to western blot data on Figure 4E, are mostly regulated by DNA-methylation, not by histone modifications (Castro-Diaz, 2014, Sanchez-Luque et al., 2019, etc). Thus, interfering with DNA methylation might be a more informative approach.

13) Experiments shown in Figure 5A have no sense at all. If MDA5 is involved in the INF response, why testing the role of DNA? Very confusing.

14) Seminal data from the Hausler lab demonstrated that ZNF93 can only regulate L1PA3 and older LINE-1 subfamilies, but not L1PA2 and L1PA1. Younger L1s are thought to be controlled mostly by DNA methylation (Castro-Diaz, 2014, Sanchez-Luque et al., 2019). Thus, the term used by authors "....overexpression of transcriptional repressors of LINE-1s (ZNF93) ..." is not accurate and it's misleading.

15) Experiments shown on Figure 5C have no controls, and authors are comparing apples and oranges. Unless authors can MEASURE the efficiency of the different shRNAs, these could not be compared. Indeed, data shown in Figure 5C contradict some of the authors findings, likely because they are comparing apple and oranges. For example, why results with shL1PA1 and shL1PA2 differ?? While shRNAs to L1PA1 seem to rescue the INF response, shRNAs to L1PA2 don't, and both are young L1s. Can authors exclude that shL1PA4 is not simply the more effective shRNA? And this is why rescue works better?? Again, that shRNAs to L1PA1 and L1PA4 can rescue but that shRNAs to L1PA2 can't is confusing and contradictory to previous data. A critical aspect here would be demonstrating that these shRNAs can reduce ds-LINE-1 RNAs.

16) Related to above, is the shMOV10 data reproducible? The SD shown is quite big. Why?

17) Data on cancer is very problematic and preliminary. Authors are exploring data collected from patients undergoing very different pharmacological treatments, and more importantly, authors are looking at the final time point, not during the generation of a tumour, which makes any conclusion indirect and likely non-reproducible. Thus, authors need to provide many more analyses to support the inclusion of these data in the manuscript. If the conclusions of this study are robust, one might expect: 1) that L1 RNAs are upregulated in the tumours that show the lower MPP8 expression, and 2) that these same tumours express markers associated with an INF response. Is this the case??? As it is, Figure 6A only supports that tumours are heterogeneous, something not surprising, but offer no direct correlation with this disease or the role of MPP8

18) The connection with DNA damage is far from been well explained, and adds nothing to the study. Or are authors proposing that indeed depressed L1s are inducing the DNA damage response as reported by Gasior and colleagues (JMB, 2006)? If this is the connection, way more support is needed.

Minor points,

a) Up to 2/3 of the human genome have been generated by Transposable Elements. DeKonig et al.2011. Please update.

b) "active LINE-1 elements with intact mRNAs" change to "active LINE-1 elements with intact

ORFs,

c) Page 5, line 114 "(ISGs) in several human cell lines (Figure 1F)." Authors should state the name of the cell lines.

d) Figure 3A, without percentages is very difficult to interpret/follow.

e) There is western blot of L1-ORF1p in HeLa in Figure 4 never referred to in the paper.

f) Figure 5C, I found surprising that only shL1PA4 is statistically significant, especially when looking at shL1PA1.

g) Warckoky et al., 2018 should be cited as a further proof that MOV10 indeed can sequester the L1 RNA.

Reviewer #3 (Remarks to the Author):

In this manuscript, the authors perform knock down of HUSH components in cultured cells (293, HeLa, etc) and demonstrate a type I IFN response (using reporter genes and gene expression profiles). The authors then utilize knockdown of putative nucleic acid sensing pathways to demonstrate that the HUSH KD induced inflammatory response is primarily MAVs and MDA5-dependent. Using RNA-seq, they observe little change in the expression of retrotransposon families but find a handful of loci (41) that are upregulated, nearly half displaying some evidence of bidirectional transcription. The authors finally show that suppressing LINEs and SVAs with specific exogenous KZNFs/KAP1 or via shRNAs targeting L1PA4 could abrogate the IFN response.

Overall the study is generally well performed, however I am not entirely convinced that LINE-1s are responsible for the effect due some experimental issues described below, and due to the well documented findings that inverted ALUs are the molecules that mediate MDA5 sensing. The manuscript would be greatly improved with additional evidence indicating LINE-mediated dsRNA is mediating the sensing by MDA5, or by showing that LINE-mediated dsRNA is sufficient to activate the MDA5/IFN pathway.

Major issues

1. In Fig 1D the authors show that the HUSH complex component TASOR does not evoke strong expression of the IFN-reporter, but instead seems to block the periphilin-mediated response in a double knockdown. The authors do not perform this experiment for TASOR and MPP8 despite MPP8 being the major component analyzed in the remainder of the study. It would be informative to know whether TASOR has the same blocking effect for the response in absence of MPP8.
2. In Fig 3D MDA5 protein levels seem elevated compared to WT cells: could this be compensation of RIG-1 absence by increased MDA5 levels? The authors could discuss this possibility. The RIG-I blot has very high background making it difficult to make out whether there is a RIG-I band in the MDA5^{-/-} cells.
3. There seems a discrepancy in whether LINE-1s are activated in response to HUSH KD. On one hand (Figure S2A) there is almost no change in the bulk expression of LINE-1s, and yet there are two figures that show that Orf1p is expressed several fold more (although it is not quantified, and frankly the blots are rather low quality, figure 4). Are any of the individual LINE-1 loci that are reactivated full length and/or fully coding for LINE-1 Orf1P? This is the only possible explanation I can see and this isn't addressed.
4. Figure 4B is and the results that discuss this figure are entirely speculative. The authors provide no evidence that the upregulation of these endogenous KZNFs play a direct role in suppressing LINE1s in response to HUSH KD. These should be removed or evidence should be provided.
5. Line 198 is misleading. The authors show that TSA induces young LINEs, which has been described before, and furthermore that TSA induces a type I interferon response (which has also been described). They in no way demonstrate that the young LINEs are sufficient for the ISG response. Comparison of gene expression changes with an unpublished dataset with engineered LINEs (without any description, lines 205-208)) is not acceptable evidence.

6. The most convincing experiments demonstrating a causative role for LINE-1s in the IFN response evoked by MPP8 KD are the overexpression of exogenous KZNFs and shRNAs targeting LIPAs. But these experiments are only done using the reporter gene. The authors should show some RT-PCR of ISGs under these conditions.

Minor issues.

1. Line 93: Later in the text an abbreviation for periphilin is used. Add this abbreviation (PPHLN1) after the first occurrence of periphilin, which we think is here.

2. Line 97: verified: add: 'by Western blot'

3. Line 114 'several human cell lines' is misleading as only one additional cell line is tested. Using instead 'in one additional cell line' would be more accurate.

4. Line 133: '7 were involved in immunity' : which 7 genes? No information given. Could add gene list to Fig 2D as for ZNFs?

5. Line 134: 'known function': please cite a reference.

6. Line 149/150: THP-1 and STING: abbreviation for ?

7. Line 151: a viral protein vFLIP: either bracket (vFLIP) or 'the' viral protein vFLIP

8. Line 178: interestingly

9. Line 186: LINEs and SINEs (16 band 14 loci respectively): inverse order: we found 14 significantly and >1 log2FC upregulated L1 loci in your supplementary table.

10. Line 205: Ardeljan et al., unpublished, while Line 604 states Ardeljan et al., in press. Please decide and use same in both lines.

11. Line 212: 'we wanted to demonstrate'. 'We wanted to test whether' would be more accurate as the outcome of the experiment was not known before starting it?

12. Line 221-222 and Fig5B: please indicate whether ZNF93 and KAP1 are significantly different from the control in the figure. The text states that both ZNF91 and ZNF93 and Kap1 can inhibit the response, but only ZNF91 is indicated as statistically significant in the figure.

13. Line 275: Most interestingly, though: maybe replace by 'It is tempting to speculate that ...)

14. Line 355: See supplementary Table 1 for sgRNA sequences. Not necessary, as already referenced in line 341.

15. Figures 1 C and S1 A: four shaded squares indicate an increasing concentration of shRNA. No information on the doses is given in the figure or the text nor methods. Please indicate this in the figure. You could e.g. have a legend with four boxes and the doses next to each box at the right side of the figure.

16. Fig 2D: the box surrounding the Venn diagram is unnecessary.

17. Fig S2 A left panel and line 177: the LINE expression change seems not to be statistically significant across the three experiments and thus the statement in the text 'transiently upregulated' might be misleading. Maybe say 'global LINE-1 RNAs were not significantly upregulated in MMP8-depleted cells.'

18. Fig 4A: intensity, not intensity.

19. Fig 4D, S1B, S2B, S3 and S4: the scale indicating the display cut-off for individual tracks in the top left corner is invisible (also in the digital file). Please label. Same applies to Gene names and chromosome location. To simplify these screenshots, the authors could write the exact genomic coordinates of the depicted locus instead of the illegible chromosome pictogram.

20. Fig4F: L1PA2 mRNA is assessed. The supplement table 2 does however, not contain a L1PA2-specific qRT-PCR primer. There is a L1PA2-3 primer. Please add missing primer or change label if 2-3 was used.

21. Fig 5C: shL1PA10-15: correct to shL1PA 10, 13 and 15 as these are the shRNA sequences listed in suppl Table 1.

22. Fig 6A: the figure would benefit from adding MMP8 before FPKM values on the y axis as all values are depicting MMP8 expression.

23. Fig6B: We suggest to swap the right panel to the left as the left panel is an extended analysis of the data on the right and mentioned after the right in the text. It would be advisable to also show the IF data in HeLa cells here or in the supplemental file which is mentioned in the text but no data is shown.

24. Fig 7: What does 'silent transcription' in the legend mean? Suggestion: LINE-1 upon MPP8 knockdown does not occur in the model despite it being a central point in the paper. Although the link of LINE-1 expression and dsRNA formation is only suggested, the authors could indicate upregulated LINE-1 expression next to the arrow from MPP8 knockdown to dsRNA and maybe add a question mark to depict the speculative character. As RIG-I also contributes to the dsRNA sensing, a + RIG-I in brackets could be added below MDA5.

25. Supplementary table 2: the 'breakdown of repeat primer hits to repeat subfamilies' is not referred to in the text. Please refer to it or omit.

26. Spreadsheet 2 (Table 220453_0_supp386903_pv7j63): This table contains 621149 NAs in column G (padj). Among those NAs are 13 features (including one L1 element) that have a $\log_2FC > 1$ and a p value of < 0.05 . In our understanding each p value should have a corresponding padj? Please have a look and correct table and resulting analysis in figures and text if necessary.

The HUSH complex is a gatekeeper of type I interferon through epigenetic regulation of LINE-1s

To the reviewers,

We would like to thank the reviewers for their detailed assessment and overall positive evaluation of our manuscript and recognition of the importance of this work in the fields of transposon biology, epigenetics and immunology. In this rebuttal letter, we address all concerns of all three reviewers with our point-by-point responses indicated below. The original reviewers' comments are in black text and our responses are in red italic text.

This work is primarily concerned with linking the HUSH complex to regulation of the type I interferon response and implicating LINE-1 elements to play a role as a novel finding of broad interest. We have now obtained the following new evidence: We show that double-stranded RNAs are produced and we demonstrate that cGAS and STING are not necessary for the interferon response observed upon depletion of MPP8. Furthermore, we now show that an engineered LINE-1 construct is sufficient to induce a type I interferon response. Most importantly, we provide new data of total RNA-sequencing in HUSH-depleted primary human fibroblasts, showing that HUSH regulates interferons and inflammation in primary cells. MPP8-depletion in these cells leads to overexpression of HUSH-regulated LINE-1 elements including full-length L1HS, which produces bidirectional transcripts that could form dsRNA. We hope that the reviewers agree that these significant advances to the manuscript provide further evidence that the HUSH complex regulates the type I interferon response through its epigenetic regulation of LINE-1 elements.

Reviewers' comments with responses

Reviewer #1

Major Issues

1.) In many cases, the data presented lacks an adequate amount of rigor. A great deal of the data presented is denoted as "one representative experiment of two shown". At the bare minimum, at least 3 replicates of an assay need to be conducted and the data should be presented as an average with the error indicated. Additionally, any representative figures should be accompanied by

quantification from the 3 separate trials. This misstep is prevalent throughout the manuscript and must be rectified.

We now plot summary data instead showing individual biological replicates (minimum n=3) and error bars.

2.) Accompanying the previous point, all data should have error bars where appropriate.

See above.

3.) The authors combine data from multiple time points in 1C, 1F, and 3B. This is not appropriate and they should have replicates for a single time point.

For all data shown, we now plot summary data for a single time point.

4.) The presentation of qRT-PCR is inappropriate. Each target should be presented with its own error bars. The combining of several targets and attempting to interpret statistics off of the clustering is not a valid approach. It is not clear how many technical replicates were used per target.

Each qRT-PCR target is presented separately with the number of biological and technical replicates stated in the legend and the error bar shown.

5.) A major caveat of this work is the almost explicit use of HEK393 cells (and the occasional Hela). This makes it difficult to assess the relevance to normal cellular biology, as both the cell lines are far removed in behavior and epigenetics from primary human cells. The addition of a primary cell line to some of these experiments would greatly benefit the impact of these findings

We now include human primary foreskin fibroblasts (HFFs) as a non-transformed cell line. These cells only grow for a limited number of passages. Upon MPP8 and PPHLN1 depletion in these cells, there is a potent activation of endogenous ISGs and IFN release as measured by IFN bioassay (Fig. 1f,k and Supplementary Fig. 1). We now include RNA-seq data on HUSH-depleted HFFs in Fig. 2 in place of the previous HeLa data (now moved to Supplementary Fig. 3d).

6.) While the authors used NRTIs and demonstrated a lack of effect on the inflammation response, it has been reported that LINE1 activation stimulates both interferon α and β (De Cecco, et al. 2019). I do not necessarily doubt the role of dsRNA in the inflammation response the authors observed, but they should address this discrepancy. Analysis of cGAS/STING via western/qRT-PCR would be prudent. Additionally, the authors should provide evidence of interaction of LINE1 dsRNA with MDA-5 (immunostaining or FISH would be appropriate).

We now provide Western blot data on cGAS/STING and MDA5 and RIG-I in all cell lines used (Fig. 3a). We also now show that we can detect accumulation of dsRNA in MPP8-depleted cells using the dsRNA-specific antibodies, J2 and K1 (Fig. 3f).

7.) Related to the above point (#6), this discrepancy with earlier publications may be the result of using highly mutated cancer cell lines in this work, which may lack functional cGAS/STING pathway. Repeating the experiments in normal cells is therefore essential. Otherwise the conclusion should be toned down to apply to cancer cell lines only.

Not all the cell lines used in this study express cGAS (Fig. 3a), suggesting the response was not cGAS dependent. However, we now test if the response induced upon MPP8-depletion is cGAS/STING dependent in cells with an intact cGAS/STING pathway. We used THP-1 cells because they exhibited the highest expression of cGAS/STING (Fig. 3a). We use knockout cell lines to show that in THP-1s, cGAS and STING are not necessary for the ISG response induced upon MPP8-depletion, while the type 1 IFN receptor is essential (Fig. 3e, Supplementary Fig. 3e).

8.) The authors should provide data for PPHLN1 knockdown for Figures 1F-H in order to concretely claim the involvement of the HUSH complex. The reliance on MPP8 for the subsequent work would be permissible if this was provided.

This is an important point thank you: we now show that depletion of PPHLN1 leads to activation of the IFN- β luciferase reporter and secretion of type I IFN as measured by IFN bioassay. The endogenous ISG response is also blocked by the JAK/STAT inhibitor (Supplementary Fig. 1d-f). The response observed upon PPHLN1 depletion is modest compared to that of MPP8-depletion and we now include all HUSH depletions in our RNA-seq analyses allowing a side-by-side comparison of differentially-expressed genes (Fig. 2).

Additional Points

-The manuscript needs to be edited for clarity. There are many grammatical errors.

Now edited

-The authors should include more about the role and function of the HUSH complex in the introduction. Additionally, the logic in Figure 2 should be expanded upon. The authors appear to focus on the ZNFs without addressing the other 23 non-ISG genes

The introduction to the HUSH complex is now expanded on. We have now also performed a detailed gene ontology analysis on all upregulated genes. Genes most affected are those involved in immunity and inflammation (Fig. 2) but pathways involving chromatin regulation and cell cycle are also affected (Supplementary Fig. 3bc).

-In general, the authors have enough space for figures that they should not have multiple graphs under the same subfigure (See Figure 1F, 1G, 3C, 3D, 4F, Figure 5)

Now corrected.

-Initial RNAi experiments should demonstrate the effects using at least 2 separate shRNAs to the same target to assuage off target contributions.

Now added: see Supplementary Fig. 1a. We also use siRNA as an independent method to deplete MPP8 in Fig. 4A.

-Figure 1F appears to have mismatch of colors in the graph vs the legend. Also, why aren't the other ISGs assayed in the HeLa cells

Now corrected.

-Figure 3A appears to be missing the shControl + shMAVS for the ISRE-GFP set

We have now coloured this overlay pink to make it more visible (Fig. 3b). Note that we also add percentages of GFP +ve cells for all samples in the key in answer to a comment of reviewer 2.

-Figure 3C lower error bar missing on first two samples

Now corrected (see Fig. 3c).

-Figure 3D western for Rig-1 is not publishable. Please get a clearer image.

We have now made new Western samples and re-run all the Westerns (see new Fig. 3d).

-Figure 3D expression is not elaborated on in Figure legend. Is this qRT-PCR data or quantification of the western?

Now removed as the new Western data is clear (Fig. 3d).

-Figure 4A is missing a box

Now corrected (see new Fig. 4a).

-Figure 4E needs to be quantified. Additionally, there needs to be an explanation for the discrepancy in the ORF1p size between the HEK293 and HeLa cells (one is ~50kd and the other is ~37kd)

This data is now quantified (see new Fig. 4b). The ORF1p size lies between the 35kD and 55kD size markers for both HeLa and 293. However, we now show only the HeLa Western (Fig. 4b).

-Figure 4F qRT-PCR targets need to be separated. Combining different targets for analysis is inappropriate

This has now been removed due to the other reviewers' finding the TSA data not to add anything.

-Figure 4G. Wouldn't the authors expect an overlap between these two data sets to be larger? There

should be some quantification and statistical statement on whether this level of overlap is even relevant. Additionally, the unpublished data should be included in this manuscript if it is to be used. *We have now quantified the overlap with L1-induced genes to be significant compared to overlaps obtained from 1000 randomizations of the same number of genes as the MPP8-repressed genes (see new Fig. 5d). Note that we now use our new RNAseq data from HFFs for these analyses and the public data we use of L1-induced genes is from a manuscript that is now published. See the legend and methods for details of the L1 construct.*

- γ H2AX foci are typically quantified. Figure 6B should be averaged counts from cells, not an intensity plot. Intensity is not reliable way of quantifying DNA damage for γ H2AX

γ H2AX-staining data are now plotted as the number of foci / nucleus (see Fig. 6c).

Reviewer #2

Major points

1) The role of TASOR. Authors suggest that TASOR depletion could rescue the effect mediated by shRNAs to periphilin, and that this could occur through activation of a negative immune regulator. Is there any evidence for this? Can author check this?

Also, is the same observed in other cellular backgrounds? As done in Figure 1F, what's the effect of TASOR shRNAs on other cell lines?

TASOR depletion can partially rescue the potent ISG response evoked upon MPP8-depletion (Fig. 1d). Note that we now perform double KD of TASOR and MPP8 instead of TASOR and PPHLN1 due to reviewer 3, comment 1. We use 293 cells, in which we are able to efficiently KD TASOR and MPP8 (Fig. 1d). We also now deplete TASOR in untransformed human primary foreskin fibroblasts (HFFs) and perform total RNA-sequencing in comparison to the other HUSH depletions (Fig. 2c-f). TASOR-depletion phenocopies MPP8 and PPHLN1 depletions in terms of affecting cell cycle and activation of TGF- α . However, TASOR-depleted cells lack an IFN response and KZNF activation. There are some up and downregulated genes unique to TASOR-depletion, listed in Fig. 2f and some of these are described in the literature to exert immunoregulatory effects as stated in the results section. However, we have deleted our previous statement suggesting that a negative immune regulator may be activated in TASOR-depleted cells because this is speculation. We now state in the text that differences in differentially-expressed genes between TASOR-depleted cells and MPP8/PPHLN1-depleted cells may indicate that TASOR required greater depletion than that achieved here (in HFFs) to observe a more complete phenotype.

2) Can authors demonstrate that the HUSH complex regulates type I INF induction on a non-transformed cell line? This is important, as karyotypic instability increase the overall copy number of TEs per cell, and what authors are reporting could be completely artefactual. At the very least, some proof of principle experiments on HCT116 (nearly diploid) should be provided.

We now provide data on three additional cell lines including untransformed HFFs (Fig. 1f, Fig. 1k) and we employ untransformed HFFs to deplete each HUSH component and perform RNA-sequencing (Fig. 2).

3) Experiments using JAK-SAT inhibitors lack proper controls, and authors should show data regarding the effectiveness of the treatment by exploring a known target.

We use IFN- β activation of the ISRE-GFP reporter as a positive control, which we verify to be blocked by the JAK/STAT inhibitor in 4 experiments (now all included in Fig. 1j). We now also show that following MPP8-depletion, mRNA expression of an endogenous ISG, CXCL10, (a known target

of IFN signaling through the JAK/STAT pathway) is also abrogated by the JAK/STAT inhibitor (Supplementary Fig. 1c) as well as ISRE-GFP reporter expression (Fig. 1j).

4) How deep is the phenotype scored in Figure 2? As the supernatant from MPP8 depleted cells is sufficient to activate an INF response (Fig 1), data shown in Fig2 correspond to a mix of cells with an activated INF response and cells responding to secreted ILs etc...thus, unless these experiments are done in a single cell manner, as presented the data is misleading, and the contribution of MPP8 could be quite small. It would be critical to conduct these assays in a karyotypically stable cell line, as much of the misregulation could be driven by increased copy number of transposons and retrotransposons.

Our new RNA-sequencing data is based in HUSH-depleted untransformed HFFs (Fig. 2). We agree that the main phenotype of upregulated ISGs will be apparent in MPP8-depleted cells and in bystander cells because IFN- β is secreted (Fig. 1g,k, Supplementary Fig. 1d) and the response is mainly dependent on the type I IFN receptor, IFNAR (Fig. 3e). We appreciate that single cell sequencing would be interesting but our main message of the ms is to connect the HUSH complex to regulation of type I IFN and underline a contributing role for LINE-1 RNAs, whereas single cell sequencing could be explored as a further mechanistic follow-up study.

5) Authors used HeLa cells to identify genes differentially regulated upon MMP8 depletion (Fig 2). Then, the RNAseq data is used to identify potentially misregulated TE RNAs, as their data suggest that the ligand driving the INF response is endogenous. As HeLa cells are known to harbour hypermethylated L1 promoters, and as their RNAseq dataset correspond to early time points, it is not surprising that LINE-1 RNAs are not much upregulated, Thus, unless additional time points are analyses, these analyses are incomplete. Indeed, and as stated above, these analyses should be ideally done in a cell with a normal karyotype, and in a single cell manner, to ensure that what authors are reporting is not linked to overall increased number of LINE and SINE elements and/or in response to secreted ILs.

Our new RNA-sequencing data in untransformed HFFs (Fig. 2) is performed at the more relevant time-point of day6 post shRNA introduction, since Initial time-course experiments revealed this time (day 6) to represent the beginning of the ISG response. We now provide a detailed analysis of LINE-1 RNAs and other TEs that are upregulated at this time-point (see Figure 4). We observe an overexpression of HUSH-bound LINE-1s.

6) Figure 4B, which is important to support the model proposed by authors (that L1 RNAs are rapidly suppressed by ZFPs), need to be expanded. As it is, there is only a very childish scheme. Although some data is shown in Figure 2D, authors need to provide compelling evidence that indeed these ZFPs can bind L1, and more importantly describe which LINE-1 subfamily is deregulated. There are many LINE subfamilies in the human genome, and this information is critical. Related to above: RNAseq analyses to identify depressed TEs should be performed using Tetrascript and/or SQUIRE, rather than an internal pipeline (there is not enough information to judge the quality of the bioinformatic approach used by authors). This is critical, as the claims are important and likely to be reproduced by others.

We have now removed Fig. 4b and accordingly, we do not suggest in the ms that upregulated ZFPs might downregulate LINE-1s since this is not the message of the ms, and indeed a different array of ZFPs are upregulated in MPP8-depleted HFFs compared to MPP8-depleted HeLa cells. RNA-seq analyses to identify derepressed TE subfamilies is performed using Tetrascript as suggested. Analyses to identify derepressed TE integrants within the genome is performed by mapping reads to the genome as would be done to identify differentially-expressed genes, except that only uniquely-

mapping reads are included, to avoid mis-assigning reads from TEs. We no longer use our previously-used internal pipeline.

7) Figure 4C: completely non informative. As above, there are >100 LINE and SINE subfamilies annotated in the human genome. Thus, specific subfamily names should be included here. This apply to LINEs, SINEs, ERVs, DNATpson and Bidirectionally transcribed TEs.

We now provide detailed information on which LINE-1 (Fig. 4d) and ERV (Fig. 4c, Supplementary Fig 4b) subfamilies and loci are derepressed in MPP8-depleted HFFs. A complete list of differentially expressed subfamilies and loci of all repeats is provided as supplementary excel files: see Spreadsheet 2: 'TE_transcripts' for all differentially expressed TE subfamilies and Spreadsheet 3: 'by_locus' for all differentially expressed TE loci. A summary pie chart of differentially expressed TE loci is displayed in Fig. 4e.

8) Authors NEVER demonstrate that indeed 5kb dsRNAs might be produced, as claimed in page 9 lanes 189-190. Thus, evidence for this should be provided. This is critical to support the model proposed by authors, which indicate that the INF response is mostly dependent on the double-stranded RNA sensor MDA5. RNA Protection Assays would be ideal for this.

We detect dsRNA using the dsRNA Abs, J2 and K1 (Fig. 3f) at day 6 post introduction of MPP8 shRNAs, at which point there is an ISG response. A time-course revealed that this dsRNA was not detectable beforehand. We have not shown in this study the identity of this dsRNA. This is why we have put a question mark in the model adjacent to 'L1 dsRNA?'(Fig. 7). We have shown, however, that the ISG response is dependent on MDA5 and RIG-1 (Fig. 3) and that at day 6, we detect a statistically significant increase in expression from both strands of full-length L1HS and L1PA2 elements in MPP8-depleted cells that have the potential to form long dsRNA of around 6kb (Fig. 4g and Supplementary Fig. 4d,e).

9) Figure 4D. These data is very problematic. On the left caption, it's quite clear that the shControl and shMPP8 generate a similar increase in transcription, from the same genomic point. Why authors state that: "Most of these instances involved LINE1 elements within genes whereby LINE1 RNA was produced in the opposite sense to the genic transcript, illustrating how double-stranded RNA of around 5kb might be produced (examples are shown in Figure 4D..." Very confusing. The example shown in the right side is consistent with the text claims, but there is no way to guess the length of the potential dsRNA generated.

We now focus on L1HS because it is bidirectionally transcribed and there is increased expression of both strands upon MPP8-depletion. Our unbiased approach to detect regions of the genome that are bidirectionally-transcribed revealed L1HS to be the top TE. Forward and reverse strands could pair in cis from the same locus or in trans to form dsRNA of around 6kb. In contrast, ancient primate-conserved derepressed LINE-1s are not scored as bidirectionally expressed but they may exhibit gene-regulatory roles since they are enriched in genes and in lncRNAs.

10) Figure 4E. It's critical to demonstrate that young L1s are derepressed in response to MPP8 depletion, as the antibody used is directed to the younger L1Hs subfamily of L1, and it's a monoclonal antibody. The data provided in Fig4C is non-informative and the two examples shown in Fig 4D (beside been problematic) correspond to evolutionary older L1s that would not be recognized by the antibody used (L1PA7 and L1M1, 30 and 61 Millions of years).

We now provide a breakdown of LINE-1s derepressed upon MPP8-depletion (Fig. 4d) and find that they span ages between 0 and 100 million years old. This includes full-length L1HS and L1PA2, for which we can detect only a modest upregulation using uniquely-mapping reads (Fig. 4h, Supplementary Fig. 5, Spreadsheet 3) but which we have found to be both upregulated by mapping to consensus as an alternative approach to avoid mappability issues (Fig. 4g, Supplementary Fig. 4d,

e).

11) Figure 4E. Don't support the claims of the authors. RNAseq analyses were done at day 3, and it's quite clear that there is no L1-ORF1p overexpression at day3. This might not be surprising, as there is a reported delay L1 in translation (see Kulpa and Moran, Hum Mol Genet 2005). Thus, a Northern blot and/or RT-qPCR should be used to really confirm that MPP8 depletion result in L1 RNA overexpression at day 3, presumably from the L1Hs subfamily according to the western data shown in Figure 4E.

L1ORF1 protein was detected at day 4 post MPP8 depletion (Fig. 4b). We now use the ORF1 protein sequence to blast translated sequences from upregulated LINE-1 loci (provided as Spreadsheet 4) and find that indeed only L1HS matches 100% suggesting that we are detecting L1HS.

12) The experiments with TSA are OVERinterpreted, and only provide indirect evidence that histone acetylation might control L1 expression (already published in reference 29). The experiment is poorly described (cell type?, concentration?, time?), but most importantly lacks control. How surprising is that TSA, a very strong IHDAC, can change the transcriptome of cells? This has been reported before, and again, there are no proper controls in this experiment. Consistently, the overlap with the Ardeljan study is minimal (16 of 77 markers). Genetic evidence should be provided here. More importantly, young L1s, which are presumably deregulated in response to MPP8 depletion according to western blot data on Figure 4E, are mostly regulated by DNA-methylation, not by histone modifications (Castro-Diaz, 2014, Sanchez-Luque et al., 2019, etc). Thus, interfering with DNA methylation might be a more informative approach.

We have removed the TSA data as reviewer 3 also found that it did not add anything to the ms. We have used randomized sets of genes of the same number as the number of upregulated genes in MPP8-depleted cells (now HFFs) to assess the overlap of these with the L1-induced genes found in the Ardeljan study. The overlap is highly significant compared to 1000 randomizations (Fig. 5d). We also now find that introduction of an engineered L1HS construct is sufficient to induce an IFN response (Fig. 5e).

13) Experiments shown in Figure 5A have no sense at all. If MDA5 is involved in the INF response, why testing the role of DNA? Very confusing.

We now show that the IFN response observed upon MPP8-depletion is not dependent on DNA sensing even in cells that express cGAS (THP-1s) in response to a comment from reviewer 1 (Fig. 3e).

14) Seminal data from the Hausler lab demonstrated that ZNF93 can only regulate L1PA3 and older LINE-1 subfamilies, but not L1PA2 and L1PA1. Younger L1s are thought to be controlled mostly by DNA methylation (Castro-Diaz, 2014, Sanchez-Luque et al., 2019). Thus, the term used by authors "...overexpression of transcriptional repressors of LINE-1s (ZNF93) ..." is not accurate and it's misleading.

This data is now Supplementary Fig. 6 because the blocking effect of ZNF93 on the response, though significant is modest and less than for ZNF91, which recognizes SVA elements. We now state that the effect of ZNF93 may be only modest because it doesn't recognize young LINE-1s such as L1HS.

15) Experiments shown on Figure 5C have no controls, and authors are comparing apples and oranges. Unless authors can MEASURE the efficiency of the different shRNAs, these could not be compared. Indeed, data shown in Figure 5C contradict some of the authors findings, likely because they are comparing apple and oranges. For example, why results with shL1PA1 and shL1PA2 differ?? While shRNAs to L1PA1 seem to rescue the INF response, shRNAs to L1PA2 don't, and both are young L1s. Can authors exclude that shL1PA4 is not simply the more effective shRNA? And

this is why rescue works better?? Again, that shRNAs to L1PA1 and L1PA4 can rescue but that shRNAs to L1PA2 can't is confusing and contradictory to previous data. A critical aspect here would be demonstrating that these shRNAs can reduce ds-LINE-1 RNAs.

We agree that the hairpins we have designed will have varying knockdown efficiencies unrelated to the age or the region of the L1 they were designed on. Efficiencies cannot be compared without recording the expression of all the LINE-1 loci that each hairpin recognizes. Our approach was instead to focus on hairpins that could block the response. We therefore name all hairpins tested a,b, c etc. except the two that could block the response, which we provide full names of in Fig.5 (see Supplementary Table 2 for full details of which sequences all the hairpins were designed on). These are L1HS (designed on the 5'UTR of L1HS) and L1PA4 (designed on L1PA4 ORF2), which recognizes 136 LINE-1s from L1HS to L1PA5 but mainly L1PA4 (47 copies). See Supplementary Table 1.

16) Related to above, is the shMOV10 data reproducible? The SD shown is quite big. Why?

The shMOV10 response is significantly higher than the control group (Fig. 5a, stats now included). Whenever there was a potent IFN response (as in the case of shMOV10), variability was high, especially as the GFP mean fluorescence intensity varies depending on which precise time-point (within day 6) we harvest the cells and catch the response.

17) Data on cancer is very problematic and preliminary. Authors are exploring data collected from patients undergoing very different pharmacological treatments, and more importantly, authors are looking at the final time point, not during the generation of a tumour, which makes any conclusion indirect and likely non-reproducible. Thus, authors need to provide many more analyses to support the inclusion of these data in the manuscript. If the conclusions of this study are robust, one might expect: 1) that L1 RNAs are upregulated in the tumours that show the lower MPP8 expression, and 2) that these same tumours express markers associated with an INF response. Is this the case??? As it is, Fig. 6a only supports that tumours are heterogeneous, something not surprising, but offer no direct correlation with this disease or the role of MPP8

Despite these cancers (TCGA data) being diverse with patients on different treatments, MPP8 is significantly downregulated in 10 out of 15 cancers (Fig. 6a). This is not the case for PPHLN1 or TASOR (now included in Supplementary Fig. 6d), suggesting that MPP8 downregulation in cancers is relevant and should be noted here. MPP8 downregulation in particular cancers warrants follow up but is beyond the scope of this study: For example, since specific LINE-1 loci are upregulated upon MPP8-depletion (Fig. 4d,) rather than global expression of LINE-1 RNAs, this point is not straightforward to address. It would need to be explored in individual cancer samples by remapping total RNA-sequencing data using only uniquely-mapping reads and could be rather an interesting follow-up study. However, we agree with the reviewer that tumours expressing lower levels of MPP8 might exhibit activation of IFN signaling as suggested by data in this study. To address this question we employ a recent work where cancers have been divided by immune subtype to help inform new hypotheses about how the immune system plays a role in the pathology and potential clearance of different cancers, independently of their tissue of origin (Thorsson et al., The Immune Landscape of Cancer, Immunity, 2018). We divide cancers by immune subtype and plot the expression levels of MPP8 mRNA in order to determine which immune subtype correlates with cancers that exhibit the lowest expression of MPP8. Interestingly, the most significant correlation is with the C2 Immune subtype that is characterized by the highest levels of IFN- γ signaling and the highest expression of ISGs, CXCL10 and CCL5 (now included in Fig. 6b).

18) The connection with DNA damage is far from been well explained, and adds nothing to the study. Or are authors proposing that indeed depressed L1s are inducing the DNA damage response as

reported by Gasior and colleagues (JMB, 2006)? If this is the connection, way more support is needed.

We now show that pathways involving DNA damage, cell cycle and chromatin maintenance are upregulated in MPP8-depleted HFFs (Supplementary Fig. 3). We explain that this coupled with the known impact of active LINE-1 elements on DNA damage and repair led us to measure DNA damage (we reference: Ardeljan, D. et al. Cell fitness screens reveal a conflict between LINE-1 retrotransposition and DNA replication. Nat Struct Mol Biol 27, 168-178 (2020); Belgnaoui et al. Human LINE-1 retrotransposon induces DNA damage and apoptosis in cancer cells. Cancer Cell Int 6, 13 (2006). Gasior, S.L. et al. The human LINE-1 retrotransposon creates DNA double-strand breaks. J Mol Biol 357, 1383-93 (2006) and Mita, P. et al. BRCA1 and S phase DNA repair pathways restrict LINE-1 retrotransposition in human cells. Nat Struct Mol Biol 27, 179-191 (2020). We state in the discussion that whether DNA damage is caused by L1s or linked to IFN-induction warrants further investigation.

Minor points,

a) Up to 2/3 of the human genome have been generated by Transposable Elements. DeKoning et al. 2011. Please update.

Now updated.

b) “active LINE-1 elements with intact mRNAs” change to “active LINE-1 elements with intact ORFs,” *Now updated.*

c) Page 5, line 114 “(ISGs) in several human cell lines (Figure 1F).” Authors should state the name of the cell lines.

Cell names now named here.

d) Figure 3A, without percentages is very difficult to interpret/follow.

Percentages are now stated in the legend (now Fig. 3b).

e) There is western blot of L1-ORF1p in HeLa in Figure 4 never referred to in the paper.

Now corrected (note this is now Fig. 4b).

f) Figure 5C, I found surprising that only shL1PA4 is statistically significant, especially when looking at shL1PA1.

Both are significant (stats now included, see Fig. 5a,b).

g) Warckoky et al., 2018 should be cited as a further proof that MOV10 indeed can sequester the L1 RNA.

Now cited.

Reviewer #3

Major issues

1. In Fig 1D the authors show that the HUSH complex component TASOR does not evoke strong expression of the IFN-reporter, but instead seems to block the periphilin-mediated response in a double knockdown. The authors do not perform this experiment for TASOR and MPP8 despite MPP8

being the major component analyzed in the remainder of the study. It would be informative to know whether TASOR has the same blocking effect for the response in absence of MPP8.

We now perform this experiment for TASOR plus MPP8 instead, which we agree is more relevant to the study (Fig. 1d) and TASOR has a partial blocking effect.

2. In Fig 3D MDA5 protein levels seem elevated compared to WT cells: could this be compensation of RIG-1 absence by increased MD5 levels? The authors could discuss this possibility. The RIG-I blot has very high background making it difficult to make out whether there is a RIG-I band in the MDA5-/- cells.

We have now re-done all the Western blots for Fig. 3d, using new samples and results are now clear. MDA5 protein is indeed elevated in the RIG-I KO compared to WT and RIG-I protein is decreased in MDA5 KO compared to WT, which we now state in the results for this section.

3. There seems a discrepancy in whether LINE-1s are activated in response to HUSH KD. On one hand (Figure S2A) there is almost no change in the bulk expression of LINE-1s, and yet there are two figures that show that Orf1p is expressed several fold more (although it is not quantified, and frankly the blots are rather low quality, figure 4). Are any of the individual LINE-1 loci that are reactivated full length and/or fully coding for LINE-1 Orf1P? This is the only possible explanation I can see and this isn't addressed.

Although specific LINE-1 loci are reactivated in response to MPP8 KD, not all loci within a subfamily are overexpressed consistent with a previous report, suggesting that LINE-1 derepression is context-dependent (Liu et al. Nature 2018). An increase in LINE-1 ORF1 protein is detectable (Fig. 4b, now quantified). This signal probably represents L1HS: The L1 ORF1p Ab was derived from the ORF1 protein sequence, which is encoded with a 100% identity to the consensus sequence of L1HS. We have been able to detect an upregulation of L1HS by mapping RNA-seq reads to its consensus sequence. Note that due to mappability issues of L1HS, we were unable to identify which L1HS loci are HUSH-regulated here. Of note, we found 199 L1HS in the genome that are expressed bidirectionally and these were full length (around 6kb) so could potentially code for ORF1 (see Supplementary Spreadsheet 4).

4. Figure 4B is and the results that discuss this figure are entirely speculative. The authors provide no evidence that the upregulation of these endogenous KZNFs play a direct role in suppressing LINE1s in response to HUSH KD. These should be removed or evidence should be provided.

Now removed.

5. Line 198 is misleading. The authors show that TSA induces young LINEs, which has been described before, and furthermore that TSA induces a type I interferon response (which has also been described). They in no way demonstrate that the young LINEs are sufficient for the ISG response. Comparison of gene expression changes with an unpublished dataset with engineered LINEs (without any description, lines 205-208)) is not acceptable evidence.

TSA data is now removed since reviewer 2 also considered that it did not add anything to the ms. For the comparison of genes induced using an engineered LINE-1 reporter and genes upregulated upon MPP8-depletion in this study (now Fig. 5d), we now provide more information in the legend and methods. We additionally show that the overlap is significant when compared to randomized gene sets (Fig. 5d).

6. The most convincing experiments demonstrating a causative role for LINE-1s in the IFN response evoked by MPP8 KD are the overexpression of exogenous KZNFs and shRNAs targeting LIPAs. But these experiments are only done using the reporter gene. The authors should show some RT-PCR of ISGs under these conditions.

We have now repeated experiments targeting L1PAs using shRNAs and looked at the endogenous ISG response for the two hairpins that could block the IFN response (Fig. 5b). Additionally, we add new data showing that expression of an engineered L1HS reporter is sufficient to activate an IFN response (Fig. 5e).

Minor issues.

1. Line 93: Later in the text an abbreviation for periphilin is used. Add this abbreviation (PPHLN1) after the first occurrence of periphilin, which we think is here.

Now corrected

2. Line 97: verified: add: 'by Western blot'

Now corrected

3. Line 114 'several human cell lines' is misleading as only one additional cell line is tested. Using instead 'in one additional cell line' would be more accurate.

We now use three additional cell lines and list them.

4. Line 133: '7 were involved in immunity' : which 7 genes? No information given. Could add gene list to Fig 2D as for ZNFs?

Our RNA-seq data is now based in HFFs and we provide detailed gene ontology information on upregulated genes in Supplementary Fig. 3. All differentially-expressed genes are also given in Supplementary data (see Spreadsheet 1)

5. Line 134: 'known function': please cite a reference.

Now cited.

6. Line 149/150: THP-1 and STING: abbreviation for ?

The full names of 'THP-1' cells and 'STING' are now stated at their first mention.

7. Line 151: a viral protein vFLIP: either bracket (vFLIP) or 'the' viral protein vFLIP

Now removed following a comment from reviewer 1. See new Fig. 3e.

8. Line 178: interestingly

Now corrected

9. Line 186: LINEs and SINEs (16 band 14 loci respectively): inverse order: we found 14 significantly and >1 log₂FC upregulated L1 loci in your supplementary table.

We now report HFF data instead (still mainly LINEs and SINEs)

10. Line 205: Ardeljan et al., unpublished, while Line 604 states Ardeljan et al., in press. Please decide and use same in both lines.

This reference is now properly cited as published.

11. Line 212: 'we wanted to demonstrate'. 'We wanted to test whether' would be more accurate as the outcome of the experiment was not known before starting it?

Now changed.

12. Line 221-222 and Fig5B: please indicate whether ZNF93 and KAP1 are significantly different from the control in the figure. The text states that both ZNF91 and ZNF93 and Kap1 can inhibit the response, but only ZNF91 is indicated as statistically significant in the figure.

Now corrected with significance indicated in the figure. Results with ZNF93 and ZNF91 are significant but KAP1 did not reach significance (see Supplementary Fig. 6c).

13. Line 275: Most interestingly, though: maybe replace by 'It is tempting to speculate that ...)

Now changed.

14. Line 355: See supplementary Table 1 for sgRNA sequences. Not necessary, as already referenced in line 341.

Now removed.

15. Figures 1 C and S1 A: four shaded squares indicate an increasing concentration of shRNA. No information on the doses is given in the figure or the text nor methods. Please indicate this in the figure. You could e.g. have a legend with four boxes and the doses next to each box at the right side of the figure.

This is now removed due to a comment from reviewer 1: we now represent one time-point (day 6) and one dose.

16. Fig 2D: the box surrounding the Venn diagram is unnecessary.

Now removed.

17. Fig S2 A left panel and line 177: the LINE expression change seems not to be statistically significant across the three experiments and thus the statement in the text 'transiently upregulated' might be misleading. Maybe say 'global LINE-1 RNAs were not significantly upregulated in MMP8-depleted cells.'

We now provide detailed information on which LINE-1 subfamilies and loci are significantly differentially expressed using Tetrascripts and using uniquely-mapping reads, respectively.

18. Fig 4A: intensity, not intensity.

Now corrected.

19. Fig 4D, S1B, S2B, S3 and S4: the scale indicating the display cut-off for individual tracks in the top left corner is invisible (also in the digital file). Please label. Same applies to Gene names and chromosome location. To simplify these screenshots, the authors could write the exact genomic coordinates of the depicted locus instead of the illegible chromosome pictogram.

Track scales are now bigger and coordinates are given above each screenshot (Fig. 4h,i, Supplementary Fig. 2, Supplementary Fig. 5).

20. Fig4F: L1PA2 mRNA is assessed. The supplement table 2 does however, not contain a L1PA2-specific qRT-PCR primer. There is a L1PA2-3 primer. Please add missing primer or change label if 2-3 was used.

We have now corrected all use of this primer to L1PA2-3 (Fig. 5c).

21. Fig 5C: shL1PA10-15: correct to shL1PA 10, 13 and 15 as these are the shRNA sequences listed in suppl Table 1.

Following a comment from reviewer 2 (comment 15) and since these shRNAs did not abrogate the IFN response which may relate to KD efficiency, we have now labelled them as d, e and f. We cannot conclude from no effect of these hairpins that there is no effect of the LINE-1s that they are targeted to. Full names are given in the Figure for hairpins that showed an effect (Fig. 5a). However, in Supplementary Table 1, information is given on which LINE-1 sequences were used to design all of the hairpins as well as their sequences.

22. Fig 6A: the figure would benefit from adding MMP8 before FPKM values on the y axis as all values are depicting MMP8 expression.

Now changed.

23. Fig6B: We suggest to swap the right panel to the left as the left panel is an extended analysis of the data on the right and mentioned after the right in the text. It would be advisable to also show the IF data in HeLa cells here or in the supplemental file which is mentioned in the text but no data is shown.

Now done and HeLa data is now in supplementary data (Supplementary Fig. 6d).

24. Fig 7: What does 'silent transcription' in the legend mean? Suggestion: LINE-1 upon MPP8 knockdown does not occur in the model despite it being a central point in the paper. Although the link of LINE-1 expression and dsRNA formation is only suggested, the authors could indicate upregulated

LINE-1 expression next to the arrow from MPP8 knockdown to dsRNA and maybe add a question mark to depict the speculative character. As RIG-I also contributes to the dsRNA sensing, a + RIG-I in brackets could be added below MDA5.

We now correct 'silent transcription' to 'Transcription OFF/low'. We now add 'L1 double-stranded RNA?' to the model and include RIG-I because it does indeed play a role.

25. Supplementary table 2: the 'breakdown of repeat primer hits to repeat subfamilies' is not referred to in the text. Please refer to it or omit.

Now referred to.

26. Spreadsheet 2 (Table 220453_0_supp386903_pv7j63): This table contains 621149 NAs in column G (padj). Among those NAs are 13 features (including one L1 element) that have a $\log_2FC > 1$ and a p value of < 0.05 . In our understanding each p value should have a corresponding padj?

Please have a look and correct table and resulting analysis in figures and text if necessary.

We have now removed NA values because these represent rows that did not meet the expression cutoff to have an adjusted p value in the DESeq2 software.

We hope that the reviewers consider our manuscript now suitable for publication in *Nature Communications* and we thank you in advance for your kind consideration.

Best wishes,

Helen M Rowe

REVIEWERS' COMMENTS:

Reviewer #1 (Remarks to the Author):

The manuscript is greatly improved from its previous version. While it is a great improvement in the right direction, I have a few lingering issues that I would like addressed prior to publication. Primarily, there are a few discrepancies in what the authors describe and what the data presented shows. Additionally, there are a few issues that require clarification.

1.) Authors claim that cGAS is not upregulated when HUSH is depleted in all cells lines that had an IFN induction upon HUSH suppression. However, TLP-1 cells show up regulation in Fig 3a. Please acknowledge this in the manuscript and address it in the discussion.

2.) Figure 5c, can the shL1HS hairpin not reduce L1HS RNA? If that is so, how do the authors explain 5a and 5b? Target qRT-PCR data should be provided for all shRNAs. This must be addressed in order for the additional conclusions in Figure 5 to stand up to scrutiny.

Other issues:

-Figure 1B bar graph needs x-axis labels

-Figure 3d shows high IFIT1 protein levels across all cell lines, but 3c indicates lower mRNA. Authors should explain this."

Reviewer #3 (Remarks to the Author):

The authors have made efforts to adequately address the majority of my concerns by performing additional experiments and/or dampening down many of the more speculative claims.

The HUSH complex is a gatekeeper of type I interferon through epigenetic regulation of LINE-1s

To the reviewers and the editorial team,

We would like to thank the reviewers and editors for their careful re-assessment of our manuscript. In this rebuttal letter, we provide a point-by-point response to the remaining reviewer queries. Queries are in black text and our responses are in blue.

Reviewer #1

The manuscript is greatly improved from its previous version. While it is a great improvement in the right direction, I have a few lingering issues that I would like addressed prior to publication. Primarily, there are a few discrepancies in what the authors describe and what the data presented shows. Additionally, there are a few issues that require clarification.

1.) Authors claim that cGAS is not upregulated when HUSH is depleted in all cells lines that had an IFN induction upon HUSH suppression. However, TLP-1 cells show up regulation in Fig 3a. Please acknowledge this in the manuscript and address it in the discussion.

We thank the reviewer for this helpful comment and we now further clarify our results regarding the contribution of cGAS/STING (DNA sensing) and RIG-I/MDA5 (RNA sensing) to the interferon induction observed upon HUSH-depletion: In Fig. 3a, we blot for cGAS +/- IFN-beta treatment not +/- HUSH-depletion. We performed these experiments to ascertain which of the cell lines used in our study could express detectable levels of cGAS, in response to a previous reviewer comment. THP-1 cells exhibit abundant expression of cGAS, which increases slightly upon IFN-beta treatment (Fig. 3a). We now highlight "IFN-β" in blue in Fig. 3a and Fig. 3d to clarify that we are treating cells with type I IFN here rather than depleting HUSH complex components. We now modify the related results section to state that cGAS was readily expressed in THP-1 cells, in which it is upregulated upon IFN-beta treatment (Fig. 3a). We agree that it is an important point that while we show that cGAS/STING are not necessary for the IFN response in MPP8-depleted THP-1 cells (Fig. 3e, as demonstrated using cGAS/STING knockout THP-1 cell lines), this pathway may contribute to the response, particularly *in vivo*. We therefore now add to the discussion: 'Of note, although the cGAS/STING pathway was not necessary for the IFN response here, it may contribute to IFN release *in vivo* and could be upregulated and more active in HUSH-depleted cells'.

2.) Figure 5c, can the shL1HS hairpin not reduce L1HS RNA? If that is so, how do the authors explain

5a and 5b? Target qRT-PCR data should be provided for all shRNAs. This must be addressed in order for the additional conclusions in Figure 5 to stand up to scrutiny.

The L1HS hairpin recognizes L1HS (L1PA1) and L1PA2 equally. Specifically, it recognizes 584 LINE1 integrants of which 288 are L1PA1 and 271 are L1PA2 and we now state this in the manuscript. We also relabel the L1HS hairpin to 'L1PA1-2' in Figure 5. Our 'L1PA2-3' qRT-PCR primers detect mainly L1PA2s (see Supplementary Table 2), including 76 L1PA2 integrants that are targeted by the L1PA1-2 hairpin. Reassuringly, with this primer set and in the presence of the shL1PA1-2, we can detect a reduction in the expression of young LINE-1s observed upon MPP8-depletion. We now include this data in Supplementary Fig. 6d and refer to it in the corresponding results section. Importantly, in Figure 4, we found that both L1PA1 and L1PA2 make bidirectional transcripts, suggesting that both of these L1 families could produce dsRNAs. Hence, it is relevant that with our L1PA1-2 hairpin, we have been able to reduce upregulation of L1PA2s. Since L1PA1-2s are restricted to the hominid rather than human lineage, we have now updated our abstract to specify that we detect bidirectional RNAs from hominid-specific L1s. Note that it is extremely difficult to design sets of qRT-PCR primers that are specific to the same LINE-1 integrants that each of the LINE-1 hairpins target. For this reason, we have focused on the two hairpins that were able to reduce the IFN response observed upon MPP8-depletion. For both of these hairpins (shL1PA1-2 and shL1PA4) we were able to detect a reduction in the expression of young LINE-1s as shown in Fig.5c and Supplementary Fig. 6d.

Other issues:

-Figure 1B bar graph needs x-axis labels
Now added thank you.

-Figure 3d shows high IFIT1 protein levels across all cell lines, but 3c indicates lower mRNA. Authors should explain this."

We clarify that Figure 3d shows IFIT1 protein expression in all cell lines after the addition of IFN-beta, while Figure 3c shows IFIT1 mRNA expression following depletion of MPP8. We now make this clearer by highlighting 'IFN-β' in blue in Fig. 3d. The results as they stand are in line with RIG-I/MDA5 knockout cell lines being able to respond to IFN-beta treatment as expected, which acts downstream of nucleic acid sensing, while not being able to mount a response to MPP8-depletion. This supports a role for RNA-sensing in the mechanism involved in IFN induction following depletion of MPP8. We now clarify in the manuscript that IFN-beta treatment of the 4 cell lines employed in Fig.c-d is an important positive control by stating: 'Importantly, we verified that all wildtype and knockout cell lines could mount a response to IFN-β treatment by measuring induction of IFIT1 and ISG15 (Fig. 3d)'.

We hope that with these text and Figure changes to reflect the outstanding queries of ref#1 and with the addition of the new supplementary data outlined above, our manuscript will be ready for publication in *Nature Communications* and we thank you in advance for your kind consideration.

Best wishes,

Helen M Rowe